# A Detailed Catalogue of Multi-Omics Methodologies for Identification of Putative Biomarkers and Causal Molecular Networks in Translational Cancer Research

**DOI:** 10.3390/ijms22062822

**Published:** 2021-03-10

**Authors:** Efstathios Iason Vlachavas, Jonas Bohn, Frank Ückert, Sylvia Nürnberg

**Affiliations:** 1Medical Informatics for Translational Oncology, German Cancer Research Center (DKFZ), 69120 Heidelberg, Germany; j.bohn@dkfz-heidelberg.de (J.B.); f.ueckert@dkfz-heidelberg.de (F.Ü.); 2Applied Medical Informatics, University Hospital Hamburg-Eppendorf, 20251 Hamburg, Germany

**Keywords:** translational cancer research, oncology, multi-omics data integration, supervised data integration, unsupervised data integration, integrative methods, analysis tools, literature review, personalized medicine

## Abstract

Recent advances in sequencing and biotechnological methodologies have led to the generation of large volumes of molecular data of different omics layers, such as genomics, transcriptomics, proteomics and metabolomics. Integration of these data with clinical information provides new opportunities to discover how perturbations in biological processes lead to disease. Using data-driven approaches for the integration and interpretation of multi-omics data could stably identify links between structural and functional information and propose causal molecular networks with potential impact on cancer pathophysiology. This knowledge can then be used to improve disease diagnosis, prognosis, prevention, and therapy. This review will summarize and categorize the most current computational methodologies and tools for integration of distinct molecular layers in the context of translational cancer research and personalized therapy. Additionally, the bioinformatics tools Multi-Omics Factor Analysis (MOFA) and netDX will be tested using omics data from public cancer resources, to assess their overall robustness, provide reproducible workflows for gaining biological knowledge from multi-omics data, and to comprehensively understand the significantly perturbed biological entities in distinct cancer types. We show that the performed supervised and unsupervised analyses result in meaningful and novel findings.

## 1. Introduction

The last two decades can be characterized as the “Post Genomic Era”, moving from hypothesis-driven approaches based on molecular and cellular methodologies (i.e., functional assays, genetic modifications of mice, animal modeling etc.) to discovery-driven approaches with the emergence of high-throughput methodologies and the area of functional genomics. Functional genomics is a field of molecular biology, which aims to understand the dynamic relationships between an organism’s genome and its phenotype, by applying different omics technologies that utilize the continuously growing body of sequence information. The term omics describes a comprehensive quantitative characterization of a class of molecules in a given biological sample or specimen, aiming to understand the molecular mechanisms and underpinnings underlying the functioning of an organism [1,2]. Currently, there are numerous single omics approaches, investigating how distinct molecular layers contribute to the manifestation and progression of various diseases [3]. Table 1 below shows an overview of the most relevant omics data types used in translational cancer research.

In the context of translational cancer research, high-throughput methodologies—and more recently the wave of NGS technologies—have highlighted significant genomic alterations in distinct solid tumors, and proposed perturbed molecular networks with potential impact on cancer pathophysiology [11,12,13]. Mutations in oncogenes and tumor suppressor genes, copy number alterations and other genetic aberrations, along with epigenomic modifications all contribute to the alteration of gene expression programs, the perturbation of normal cellular processes and the promotion of tumor formation [14]. Understanding these biological processes may enable the development of novel therapeutics and the faster detection of various types of cancers [15]. One representative example of published studies using cancer genomic data on a global scale is The Cancer Genome Atlas (TCGA) consortium, a landmark cancer genomics program funded by the National Cancer Institute in 2006 [16], and its Pan-Cancer Atlas initiative. The Pan-Cancer project is the largest and most comprehensive molecular analysis of multi-omics sequencing data and clinical annotation from more than 10,000 samples, comprising 33 of the most prevalent forms of cancer. In detail, the computational analysis with collectively 27 publications led to the identification of 299 cancer-driver genes and over 3400 driver mutations. These results shed light on the molecular underpinnings of cancer, such as cell-of-origin patterns and oncogenic processes, which classify distinct solid tumors and could serve as a valuable resource for precision medicine [17].

### 1.1. Limitations of Single-Omics Approaches in Complex Phenotypes

The majority of diseases and human disorders have extremely complex phenotypes, with confounding variables making it difficult to detect a clear causality [18]. Similarly, in the vast majority of cancer types studied through single-omics high-throughput experiments, the hidden biological variation can be represented as the metaphorical tip of the iceberg. For instance, extracting a list of differentially expressed genes, somatic point mutations or copy number alterations provides a limited understanding of the studied malignancy, not reflecting the total molecular complexity [19]. Additionally, various biases associated with each technology—based on both analytical and statistical thresholds and related to the experimental design—further confound each separate bioinformatics analysis [20]. Therefore, by interrogating only a single-omics experiment, one cannot identify the interplay between the different molecular entities, and thus unravel the causal mechanisms that comprehensively describe the diverse nature of each cancer. Ignoring the complexity of underlying molecular mechanisms could lead to wrong assumptions or misinterpretation of results.

### 1.2. Multi-Omics Concept Introduction and Background

The multi-omics analysis approach follows the core principle that any biological condition or disease such as cancer constitutes multiple molecular phenomena, and only through the detailed understanding of the interactions between the different molecular layers can one understand holistically the significantly perturbed biological entities that characterize the specific disease [21,22].

These approaches have been employed to predict vaccine response [23] or to link complex phenotypes with multi-omics profiles in genome-wide association studies (GWAS) [24,25].

Personalized medicine can benefit from the implementation of multi-omics data integration methods, as the profound diversity in disease onset, progression and treatment outcome across cancer patients makes it difficult to decide on the optimal patient-specific treatment. Thus, joint analysis of multiple omics layers (‘multi-view learning’) may lead to a better understanding of heterogeneity and thus personalized treatment decisions [26,27,28]. Furthermore, single-cell multi-omics approaches can help to disentangle the different factors contributing to cell-to-cell heterogeneity [29], and therefore build a more complete snapshot of cancer biology, especially with respect to tumor clonality and treatment relapse [30].

### 1.3. Public Cancer Multi-Omics Data Repositories

As multi-omics approaches require high-dimensional data, portals hosting these data have high standards for normalization and transparent harmonization of different molecular omics modalities. In this section we would like to highlight different data repositories for multi-omics purposes and their utility for translational cancer research. For a more detailed and broader overview on leveraging distinct omics databases for personalized oncology see [31].GDC: The Genomic Data Commons Data Portal (https://portal.gdc.cancer.gov/) from the National Cancer Institute (NCI) is the largest scale consortium that enables the retrieval, download, comprehensive analysis and exploitation of multimodal cancer genomics studies. Within the GDC, scientists can access over 3 petabytes of data from programs like the NCI’s Clinical Proteomic Tumor Analysis Consortium (CPTAC), the Therapeutically Applicable Research to Generate Effective Treatments (TARGET) initiative and The Cancer Genome Atlas (TCGA). Both harmonized datasets and legacy data on older genome versions are available. Furthermore, it includes various data visualization tools to enhance the exploration of specific projects and cancer types, and available bioinformatics pipelines. Based on the latest data summary (October 27, 2020), this data-driven platform contains 67 projects, 68 different cancer types with more than 84 thousand cases, along with clinical data [32].ICGC: The International Cancer Genome Consortium was established in 2008 as an international effort to harmonize the large number of ongoing and future projects on cancer genomics. Members include the NIH, the Wellcome Trust Sanger Institute, Cancer Research UK, RIKEN, and many more. Its data portal (https://dcc.icgc.org/) currently holds 86 projects with more than 80 million somatic mutations (data release 28). Its flagship projects are the Pan-Cancer Analysis of Whole Genomes (PCAWG) and ARGO (Accelerating Research in Genomic Oncology). Launched in 2019, ARGO is the next phase of ICGC, which aims to uniformly analyze specimens from 100,000 cancer patients with high quality clinical data to address outstanding questions in genomic cancer research (https://www.icgc-argo.org/).PCAWG: The Pan-Cancer Analysis of Whole Genomes is the latest ICGC initiative with data released in 2020, and one of the biggest international collaborative studies, including 13 research institutes and more than 700 scientists from individual TCGA and ICGC working groups. The major aim of the PCAWG consortium is to identify common mutational patterns and to investigate the nature and consequences of somatic and germline mutations of 38 cancer types collected from 48 TCGA & ICGC projects. The project data is available from five major resources [33]:The ICGC Data Portal (https://dcc.icgc.org/repositories) as the main dissemination platform for ICGCUCSC Xena (https://xenabrowser.net/) is the main exploration tool for the included multi-omics resource data, in order to identify any putative correlations among all primary results. Additionally, it includes the possibility of performing survival analysesEBI Expression Atlas (https://www.ebi.ac.uk/gxa/) is an open science resource hub hosted by EMBL/EBI. It provides information about gene and protein expression across species and biological conditions such as different tissues, cell types, developmental stages and diseases.BSC PCAWG Scout (https://pcawgscout.bsc.es) is another analysis platform to visualize and explore PCAWG data. It consists of a portal that presents the original omics data and sample annotation along with the results from different analysis working-groups, whereas its main focus lies on providing information on driver mutations and resulting proteins.Chromothripsis Explorer (http://compbio.med.harvard.edu/chromothripsis/) is a tool that provides highly interactive Circos plots for all tumors in the PCAWG cohort. Each Circos plot reports the point mutations, small insertions and deletions, structural variations and copy number profiles detected in each tumor. On this premise, the user can exploit large-scale alterations such as chromosome arm deletions, and complex mutational patterns such as chromothripsis.CCLE: The Cancer Cell Line Encyclopedia (https://portals.broadinstitute.org/ccle) is an ongoing collaborative project between the Broad Institute and the Novartis Institutes for BioMedical Research. Established in 2008, its main goal is to conduct a thorough genetic and pharmacologic characterization of a large panel of 1457 cancer cell lines. Its aims are to capture the genomic heterogeneity of the preclinical models and link them with the molecular heterogeneity in cancer patients. Additionally, it can unravel clinically actionable molecular targets that might be associated with drug response and ultimately link them to cancer survival, enhancing personalized medicine. Collectively, the multi-omics cell lines dataset includes gene expression from microarray and RNA-Sequencing experiments, reverse-phase protein arrays, copy number, gene methylation and mutation data. In parallel, the database also stores legacy data, which include pharmacological profiles of 24 anticancer drugs across 504 cell lines. Besides data the web page also includes tools for data visualization, including box plots, scatter plots and bubble maps for methylation data [34].cBioPortal: The cBioPortal for Cancer Genomics (https://www.cbioportal.org/) is an open-source resource platform developed at Memorial Sloan Kettering Cancer Center, whereas the software is developed and maintained by various research institutes. Its main goal lies in the interrogation, interactive visualization and integrated analysis of clinical and complex multimodal cancer genomics datasets. While the major focus of the platform lies on genomic alterations (non-synonymous somatic mutations, DNA copy-number variations), it also hosts mRNA and microRNA expression, protein and phosphoprotein level data (RPPA or mass spectrometry based), DNA methylation and microbiome data, especially for TCGA data. Additionally to TCGA projects, cBioPortal includes other large-scale cancer genomics projects to advance translational cancer research, such as immunogenomic and pan-cancer studies. Overall, whereas cBioPortal is considered mainly as an exploratory analysis tool, GDC would be a more appropriate choice if the user requires full access to raw data from various cancer projects (TCGA, TARGET). Additionally, cBioPortal is currently using only data aligned to the hg19/GRCh37 reference genome, and it doesn’t provide normal tissue samples for any study [35,36].COSMIC: Besides cBioPortal, the Catalogue Of Somatic Mutations In Cancer (COSMIC)–developed at the Wellcome Sanger Institute—is the largest and most comprehensive resource for mining publicly available cancer sequence data, aiming to investigate the impact of somatic mutations on cancer progression and pathophysiology [37]. The latest version (COSMIC v. 92, August 2020) contains more than 37 million coding mutations and other clinical information from more than 1500 cancer types both on GRCh38 (hg38) and GRCh37 genomes. The impact of somatic variants can be summarized on various levels and across projects, such as clinical actionability (drug resistance), mutational processes associated with cancer progression (mutational signatures) and more. Furthermore, COSMIC includes the Cell Lines Project (CLP), a multidimensional dataset containing a detailed molecular characterization of more than 1000 cancer cell lines with copy number variation and gene expression data, including also other previously published moderate scale sequencing projects, such as the NCI-60 Human Tumor Cell Lines Screen dataset [38].

### 1.4. Platforms and Packages for Leveraging Multi-Omics Data Retrieval

Development of platforms and packages for accessing, configuration and preparation of data in the field of multi-omics data integration makes tools easier applicable and saves time for the major integration and analysis of data but is also limited to the use of specific runtime environments. The mainly used programming languages in this field are R and python (see following Section 2). Efforts to use a multi complex runtime environment by including both languages have led to the development of Python-R interfaces like rpy2 (https://pypi.org/project/rpy2/) and reticulate (https://rstudio.github.io/reticulate/) which have been used for multi-omics data integration, especially for combining machine learning computations and data mining approaches [39]. The main innovation and development in open source machine learning platforms like TensorFlow [40] and PyTorch [41] make python the language of choice for ML-development and related applications.

In addition, recent implementations in R led to integration of functionalities from those two big platforms to make ML-development also available for R users (https://github.com/rstudio/tensorflow, https://github.com/f0nzie/rTorch). The main source of development for bioinformatics packages in R is the Bioconductor software platform. It is an open source and open development project, which provides tools for the exploration and analysis of high-throughput omics data. It is based on the R programming language, and among its main priorities are reuse and interoperability, along with high-quality documentation. In R, the fundamental unit of sharable code is the R package, which combines code, data, tests and vignettes, which are extensive documents illustrating how to use the corresponding package. The latest version (3.12) includes 1974 packages, covering a broad range of bioinformatics and statistical applications for sequencing data (RNA-Seq, ChIP-Seq, variant annotation etc.), microarrays, flow cytometry, imaging and proteomics [42,43]. Regarding translational cancer research, there are a number of important R packages that facilitate the management, assessment and download of TCGA data from the aforementioned public data resources. These include but are not limited to GenomicDataCommons, TCGAbiolinks, cBioPortalData and curatedTCGA R packages, with varying strengths in ease-of-use, integration, and completeness of data. For example, GenomicDataCommons [44] offers full access to all available files from the TCGA and other studies. TCGAbiolinks [45] additionally reduces the burden of computational time and data processing when starting from raw or not fully transformed data, by providing a single-omics data type harmonization with the “SummarizedExperiment” data container, along with the accompanied clinical data for the selected cancer studies. Furthermore, the R package curatedTCGAData [46] aims at balancing interoperability with complexity, by offering an integrative and user-friendly representation of multimodal TCGA data for download in Bioconductor [47]. The package is based on the MultiAssayExperiment (MAE) software, an integrative representation for multi-omics data studies. MAE is a Bioconductor object-oriented S4 class general data structure, which is modelled after the “SummarizedExperiment” representation for expression data and coordinates multi-omics experiments on a set of biological specimens [48]. Moreover, MAE can incorporate any number of assays with distinct representations and dimensions. Assays have to be either “range-based” (measurements relate to genomic ranges such as gene expression or copy number) or “ID-based” (measurements are indexed identifiers of genes, proteins, microRNAs, etc.). The package curatedTCGAData can yield and construct “on the fly” MAE representations from flat files of 33 different cancer types from the Broad GDAC Firehose (hg19 data). Finally, the cBioPortalData package provides an R/Bioconductor interface to fetch, expose and utilize cBioPortal cancer data. It imports cBioPortal datasets as MultiAssayExperiment objects into Bioconductor, in order to construct integrative representations of multi-layered studies. Moreover, cBioPortalData implements two main approaches for accessing the data: one for downloading pre-packaged and another for sending queries through the cBioPortal API. One current limitation is that the user can only query specific gene panel combinations within a study.

On the other hand, python-based development in this field has also led to numerous useful tools for accessing, preprocessing, analyzing, and integrating multi-omics data from cancer repositories. Such tools like TCGAIntegrator [49], PyGDC (https://github.com/hammerlab/pygdc), xenaPython [50] and OpenOmics [51] helped by accessing APIs, prepare and integrate multi-omics data from widely used web platforms such as TCGA/GDC or cBioPortal in various studies [52,53,54]. To date, based on the Python Package Index (PyPI) repository, 197 bioinformatics related projects for multi-omics data are currently stored (https://pypi.org, accessed: 4 March 2021). Finally, besides the Bioconductor and Python projects, also other web based curated platforms provide tools to perform multi-omics data analysis. The most representative example is the online platform Galaxy [55], including various interfaces for the integrative visualization and exploration of multi-modal layers, such as the Multi-omics Visualization Platform (MVP) plugin suited for proteogenomic data analysis [56].

### 1.5. Challenges Integrating Multi-Omics Experiments

Despite the wealth of different cancer omics layers deposited in the aforementioned databases, there are some noticeable challenges regarding their efficient integration and interpretation. Firstly, one of the major bottlenecks is the multi-layered data acquisition. Heterogeneous data collected using different techniques (i.e., data modalities) generally exhibit distinct statistical properties (discrete analytical ranges), which could be attributed also to inter-patient individual genomic diversity, cell type composition and other technical factors. Additionally, this complexity is further enhanced by the inherent correlation structures and hidden confounders (i.e., systematic errors) introduced by each different omic layer [57,58,59]. A representative example is the integration of proteomics with other types of omics data, such as transcriptomics, as the former are usually investigating a limited percentage of the expressed genome, are more challenging in the experimental preparation, with additional effects (post translational modifications, localization and/or degradation) further perplexing the modeling of inter-data relationships [60]. Moreover, one other limitation lies in the absence of a “standardized” protocol for sharing and storing the available multi-omics data in the various cancer data repositories, resulting in the “under-utilization” of the available molecular information being present. In particular, different web platforms, host multi-modal cancer data in distinct processing and transformation formats (different normalization pipelines, reference genome versions). This absence of “harmonized” data containers pertains a major obstacle to researchers trying to utilize or compare different studies, or even to reproduce original findings from published initiatives. Thus, this augments the necessity for reproducible, common and standard data representations pertaining multi-omics cancer studies. Finally, another bottleneck is the presence of large amounts (and in parallel different patterns) of missing values, mainly in the clinical data, but also amongst the same patients being profiled with different high-throughput experiments. This results in sparse datasets, frequently including non-matched tumor-normal samples, missing percentages of profiled omics layers and inaccessible clinical annotations for the studied patient cohorts [39].

### 1.6. Research Outlook: Single-Cell Multimodal Analysis

Single-cell multimodal omics represents the recent technological advancement from single-cell RNA-sequencing (scRNA-seq) to the acquisition of multiple molecular data types such as genome, transcriptome, methylome or proteome from single cells.

This includes the combination of multiple next-generation sequencing-based methods such as DR-Seq (gDNA-mRNA sequencing) [61] and G&T-Seq (genome and transcriptome sequencing) [62], ATAC-RNA-Seq (combined assay for transposase-accessible chromatin using sequencing and RNA sequencing) [63] or the capture of three-dimensional genome structures with DNA methylome profiling (scMethyl-HiC [64] and snm3C-seq [65]). Additionally, droplet-based methods such as Perturb-Seq [66,67], and CRISP-Seq [68] have been developed, which combine CRISPR-based transcriptional interference with high-throughput single-cell RNA sequencing. For a full review on experimental methodologies see Zhu et al. 2020 [69], Ma et al. 2020 [70] or Lee et al. 2020 [71].

This powerful technology enables the investigation of complex biological states and processes of multicellular organisms. In cancer research it can be used to explore tumor heterogeneity, tumor evolution or the identity of infiltrating immune cells [72,73,74]. A triple omics single-cell sequencing approach in hepatocellular carcinoma for example identified two subpopulations of carcinoma cells, which significantly differed in DNA copy number, DNA methylome, and transcriptome [75]. A study in cutaneous squamous cell carcinoma combined scRNA-Seq with spatial transcriptomics and multiplexed ion beam imaging and uncovered multiple features of potential immunosuppression in the compartmentalized tumor stroma [76].

The aim of analyzing multimodal single-cell data is the unification of different data modalities to uncover complex biological mechanisms on the cellular level such as the reconstruction of gene-regulatory and signaling networks [77]. Particular challenges of this approach lie for one in the still low throughput and high cost of multimodal single-sequencing assays often leading to data sparsity. Additionally, technical noise is often high due to low sequencing coverage and missing values [69,78].

Often not all modalities of a data set stem from exactly the same cell but cells from the same sample or tissue, leading to batch effects from unmatched data. To remedy this projection into a common latent space (Feature Projection) can be applied. Canonical correlation analysis (CCV) and Manifold alignment are both feature projection-based dimensionality reduction techniques. CCV, which is a multivariate analysis technique for estimating a linear relationship between two sets of measurements, can be performed using Seurat3 [79]. VDJView [80] is a specialized tool for the multimodal analysis of data from T and B cells, and includes Seurat [81], Scater [82] and SC3 [83] as well as several additional analysis and visualization features. Manifold alignment algorithms such as MATCHER [84] or MMD-MA [85] use a type of machine learning algorithm that produces projections between sets of data, given they lie on a common manifold.

Bayesian Modeling is a stochastic variational inference method based on Bayesian modeling [86]. Clonealign [87] integrates expression and copy number data from human cancers under the paradigm that copy number is positively correlated with gene expression. BREM-SC [88] is a random effects mixture model for the joint clustering of paired single cell transcriptomic and proteomic data.

Regression Models include least absolute shrinkage and selection operator (LASSO) regression with sci-CAR [89], gradient boosting regression (GBR) modeling [90], Hidden Markov random field (HMRF) modeling with trendsceek [91] and multivariate normal modeling (MNM) with SpatialDE [92].

In addition, single-cell multimodal autoencoders for mapping to a shared latent space are emerging [93,94].

For the unsupervised integration of single-cell multimodal data a widely used method is Matrix Factorization. Here, the data matrices are decomposed into two lower dimensionality matrices. Methods include integrative non-negative matrix factorization (iNMF) by algorithms such as Wishbone [95] or LIGER [96], coupled nonnegative matrix factorization (coupleNMF) [97], group factor analysis (GFA) with algorithms such as Multi-Omics Factor Analysis (MOFA+) [98], and independent component analysis (ICA) [67].

Additionally, MIMOSCA [66] and MUSIC [99] are algorithms for the analysis of expression data after CRISPR perturbation (Perturb-Seq). While MIMOSCA is based on a regularized linear model to estimate the impact of perturbations on gene expression, MUSIC utilizes topic modeling, a decomposition method to discover the shared latent information among input matrices as used for the discovery of hidden semantic features in natural language processing.

Finally, an interesting new implementation is included in the new release of the DESeq2 R package: the function integrateWithSingleCell integrates bulk differential gene expression analysis results from DESeq2 with public single-cell datasets. This facilitates the investigation of which types of cells might be responsible for the relative expression differences in the bulk samples [100].

## 2. Results

### 2.1. Literature Mining

The general literature search in PubMed was performed in November 2020 and resulted in 753 publications about multi-omics data integration. We started our search with basic search terms to discover the entire complexity of multi-omics publications and continued with more specific filtering for supervised methods, unsupervised methods, reviews, tools, and cancer related papers. A detailed description of the literature mining parameters is available in the Appendix A. Appendix A shows the distribution of classifications of these publications. We classified 91.5% (688) of all papers, some of which also have multiple classifications as reviews may also deal with supervised or unsupervised tools in cancer research. Most papers (75% (566)) were classified in the tool category, where they either apply tools or report new tools. Only 3% (24) were classified as dealing with supervised multi-omics data integration, whereas about 18% (135) were classified as dealing with unsupervised multi-omics data integration. Figure 1 shows the overlapping distribution of the three categories Cancer, Review, and Tool for supervised, unsupervised and other not clearly classified papers. The class called “Other” in Figure 1 contains papers which are not clearly classified as supervised or unsupervised multi-omics papers but they can deal with both supervised and unsupervised, or semi-supervised data integration. Only 3% are related to supervised multi-omics data integration which can be fully classified into the overlapping subcategories Cancer, Tool, and Review whereas the subcategories for unsupervised and other papers cover 91% and 89% of search results. We observed that publications classified as tools, which is the largest subclass, have overlap with the review subclass and predominately to the cancer subclass. The dominating trend of unsupervised tools in comparison to supervised tools can also be observed by looking at published summaries of multi-omics data integration tool classifications [101,102]. Anyhow, 8.5% (64) publications did not fit into any of these categories. Additionally, Appendix A shows the distribution of classified papers according to the publication year. There we see a drastic increase of multi-omics related papers starting from 2012 to 2020 where publications dealing with tools are cover the majority of publications followed by Cancer related papers and Reviews. This displays the development in the field and the importance of multi-omics research in general and in translational cancer research. A full list of mined publications can be found in Appendix A.

### 2.2. Methodologies and Motivation

In this section, we will give a summary of user-friendly tools suited for defined general research purposes, using tools identified in our literature research as well as tools classified by Nicora et al. (2020) [103] and Huang et al. (2017) [101]. Criteria for selected tools were that they should integrate more than one omics layer, should show cancer-related use cases or demonstration on cancer data, and they should have a clear documentation for user-friendliness, such as a vignette or repository with sufficient information.

We used common general research purposes as defined by Nicora, et al. (2020) [103], but added “cancer subtype classification” to the list for cancer-specific analyses. The following Tables 2–7 reflect this classification and the wide range of developed tools available for multi-omics data integration. The categorization of these tools is based on the most common use case of each tool, which does not necessarily mean that the tool is limited to that research purpose but has been mainly applied for this aim. We also classified the tools based on criteria for supervised and unsupervised learning. For supervised learning methods, the tool has been trained on labeled training data in order to optimize a given hyperparameter for the defined hypothesis and to minimize a specific loss function. Unsupervised tools do not utilize labeled data. They can learn from non-labeled data with unknown non-categorized patterns. However, they are commonly based on statistical methods rather than on machine learning techniques. Additionally, the tools in Table 7 are used for multiple research purposes and can be applied to several research aims.

#### 2.2.1. Patient Stratification

Multi-omics data integration tools for patient stratification (see Table 2) are finding groups of samples of therapeutic and clinical relevance. These groups can be defined by a multi-omics profile for specific treatment response or survival benefit.

#### 2.2.2. Biomarker Discovery

Tools for the discovery of biomarkers (see Table 3) are aiming to find specific composite molecular signatures for clinical (prognostic and/or diagnostic) utility such as disease state, treatment response, or survival rate. Multi-Omics integrated biomarker sets can encapsulate changes and effects in different omics layers as key points for personalized medicine (e.g., mutations leading to changes in expression, protein folding, genetic regulation, or methylation). Representative examples include the derivation of composite signature sets in the field of radiogenomics and colorectal cancer [109] and in the pan-cancer classification of distinct solid tumors [110].

#### 2.2.3. Pathway Analysis

Tools for pathway analysis (see Table 4) are dealing with regulatory effects (e.g., gene regulatory networks, post-translational modifications), interactions between different pathways on multiple omics layers (e.g., gene/protein interaction networks), or use databases like KEGG [119] or REACTOME [120] for pathway discovery. Except for these, some additional and widely used resources of prior knowledge include the Molecular Signatures (MSigDB) [121] and the Pathway Commons [122] databases. MSigDB is a comprehensive resource of annotated gene-sets separated into nine major collections, whereas Pathway Commons comprises of an integrated repository spanning about 4794 biochemical processes and 2.3 million interactions. In addition, the SignaLink 2 resource [123] is a signaling pathway database with multi-layered regulatory networks for the interpretation of multi-omics studies. Finally, the Omnipath database [124] is one of the richest sources regarding protein-protein interactions, including more than 100 knowledge resources for 20,000 human proteins and 16,500 complexes.

#### 2.2.4. Drug Analysis

The following tools in Table 5 aim at the discovery of new drugs or new drug effects and the use of existing ones in combination with others for improved drug response and survival based on data from different omics layers. These tools try to identify potential drug targets in search for better treatment with higher survival rates, and are applicable in the field of pharmacogenomics and drug repurposing, where multi-omics analysis can identify putative target regulators, which affect dynamic molecular networks (e.g., drugs targeting identified pathways resulting from analysis of differences in gene and protein expression) [133].

#### 2.2.5. Cancer Subtype Classification

The tools in this category are used for the classification of molecular subtypes of specific cancer types (see Table 6).

The identification of novel disease subtypes can be improved by the application of these integrative methodologies. They can lead to the identification of improved targets for anti-cancer treatment or they could contribute additional knowledge to existing cancer subtypes from a multi-view perspective.

**Table 6 ijms-22-02822-t006:** Summary of computational tools for Cancer Subtype classification in the field of multi-omics data integration.

Model Nature Orientation	Tool Name	Programming Language	Integration Method	Used Omics	Reference
Unsupervised	mixKernel	R	Multiple Kernel learning	mRNA, miRNA, MET	[136]
Unsupervised	iClusterBayes	R	Bayesian clustering	mRNA, CNV, MUT, MET	[137]
Unsupervised	SNF	R, Matlab	Network fusion	mRNA, miRNA, MET	[138]
Unsupervised	iCluster	R	Matrix factorization	mRNA, CNV	[139]
Unsupervised	iCluster Plus	Matlab	Matrix factorization	mRNA, CNV, MUT	[140]
Unsupervised	JIVE	Matlab	Matrixfactorization	mRNA, miRNA	[141]
Unsupervised	PSDF	Matlab	Bayesian	mRNAs, CNV	[142]
Unsupervised	BCC	R	Bayesian	mRNA, miRNA, MET, PROT	[143]
Unsupervised	SCFA	R	Multi-step analysis	mRNA, miRNA, MET	[144]
Supervised	MAUI	Python	Autoencoder	mRNA, CNV, MUT	[145]

mRNA = Gene Expression, miRNA = microRNA Expression, MET = Methylation, CNV = Copy Number Variation, PROT = Proteomics, MUT = mutation.

#### 2.2.6. Multi-Omics Data Discovery

The previous research aim categories are sometimes very closely related (e.g., drug discovery implies sometimes biomarker discovery for detecting effective druggable marker). Therefore, several tools can be employed in multiple of the selected research aims (see Table 7). Most of them are unsupervised and can be employed for carrying out an initial exploratory analysis on multi-omics profiles of different cancer types.

Generally, we propose three main criteria that could guide the appropriate selection of all the above catalogued methodologies:Research aim: Firstly, the most important aspect is the research question: what is the purpose of the specific study, or which biological insights are aimed for regarding a specific cancer type or cohort? This can significantly inform the selection of the most suitable tools that match the specific research goal, as no single tool or pipeline covered above can address a complex disease like cancer in its entirety. In addition, except for the initial selection (i.e., unsupervised or supervised methodologies), benchmark studies can be further utilized and considered as guidelines for narrowing the candidate tools [156].Another crucial criterion is the experimental design and the interrogated datasets: which is the relative sample size? For example, usually ML-based approaches require a higher number of samples for model training and validation in comparison to unsupervised methodologies. Also, can the relative tool cope with the percentage of missing values and/or the nature of omics layers (continuous vs. sparse genetic data)? When a large percentage of missing values is present in both omic layers and clinical data, researchers should consider various published studies covering different methodologies for missing value imputation [157].Furthermore, the third selection criterion that is often underestimated is the presence of extensive documentation that accompanies an available tool. Despite the fact that an approach might be well suited for a specific research scenario, the absence of a rich vignette and detailed reproducible examples poses a significant constraint on the utilization of the respective methodology [158].

### 2.3. Rationale for Selection of Tools and Datasets

Overall, as already pinpointed in the literature mining process above, there is a large amount of computational tools and pipelines that can be utilized for different research goals. However, few studies or reviews provide also comprehensive examples or tutorials on how to utilize public cancer genomics repositories, and perform multi-omics data integration. On this premise, we selected two tools that can be utilized in two distinct scientific scenarios: the MOFA R package for unsupervised methodologies, and the netDx R package for the supervised ones. Initially, the main rationale of selecting R language tools is that the Bioconductor project is the largest consortium for the statistical analysis and comprehension of genomics data (https://bioconductor.org). It is comprised of a core team of more than 1200 researchers to support continuous development. It is widely used with around ¾ million distinct IP downloads annually, and well respected (42,000 PubMed Central full text citations). Moreover, it provides detailed documentation and extensive vignettes based on high quality standards and a broad scientific community that can provide support (https://support.bioconductor.org). Additionally, each package is thoroughly tested in different computational systems for scalable and performant analysis. Furthermore, the majority of the above selected tools are based on the R language.

Concerning the non-supervised approach, MOFA/MOFA+ [98,159] was chosen as the respective methodology, as it is by design unsupervised, so it is not aimed at detecting differential changes between a predefined set of samples. It provides a well-established workflow to characterize these sources of variation, especially when analyzing datasets with complex group structure. Also, MOFA+ has been extensively used and cited in more than 80 research studies and comparative reviews [160,161,162,163]. Furthermore, the MOFA+ stable Bioconductor installation is utilizing basilisk to automatically set up the necessary Python-R connection, which facilitates interoperability.

On the other hand, netDx [107,164] is a recently published Bioconductor R package, which provides a novel methodology of implementing patient similarity networks for efficient patient classification, which has been shown to outperform other machine learning approaches. It can integrate heterogeneous patient data from clinical to omics layers, while implementing machine learning algorithms for robust feature selection. Furthermore, it uses Cytoscape (RCy3) for the efficient visualization and interpretability of the inferred biological networks. Finally, as the two aforementioned methodologies can’t be directly compared, we selected two different multi-omics cancer datasets, for the different computational approaches. In particular, the TCGA-LUAD dataset was selected for the unsupervised approach, based on the absence of known molecular subtypes between the patients. For the hypothesis driven netDx approach, we selected the CLL dataset, as the IGHV mutational status is a known clinical marker that separates patients into distinct classes (see Materials and Methods Section 4.2 and Section 4.3).

### 2.4. Unsupervised Multimodal Data Integration Case Study with MOFA

In order to disentangle the heterogeneity and unravel new biological insights regarding lung adenocarcinoma, MOFA+ analysis was applied in the processed TCGA-LUAD dataset, as described in the Materials and Methods section.

An initial overview of the trained MOFA model is illustrated in Figure 2. In detail, in Figure 2A we observe the correlation between the inferred latent factors from the model, which verifies that all factors are mostly uncorrelated, suggesting a good model fit. Figure 2B shows the percentage of variance explained by each factor across each omics layer. Interestingly, Factor 1 seems to capture a source of variation that is presented across two modalities, being gene expression (RNASeq) and protein abundance (RPPAArray). In contrast, Factor 2 seems to capture a strong source of variation that can be attributed mainly to the gene expression data, whereas Factors 3 and 4 are mainly related to the CNV data. Collectively, in this dataset using in total 15 Factors, the model explained up to ~43% of the variation in the gene expression data, around 38% in the copy number alteration data, and ~18% in the RPPA data. Overall, the above findings suggest that no single omics technology can explain holistically all the sources of variation in the dataset, further augmenting the necessity of profiling a complex disease with different molecular layers.

Next, aiming to explore the molecular landscape of lung adenocarcinoma, we initially performed a correlation analysis to associate the MOFA factor values with any included clinical sample metadata. The analysis highlighted that expression subtype, ATM mutation and gender had a significant correlation (log10 adjusted *p*-value <0.05) with specific factors. In detail, visualization of the samples in the latent space showed that the expression subtype had an association with Factor 1, clearly separating the terminal respiratory unit (TRU) subtype from other two (Figure 3A), whereas ATM mutation showed an interrelation with Factor 2 (Figure 3B). Notably, ATM gene somatic mutations have been illustrated to play a role in the pathophysiology of lung cancer [165,166].

In parallel, for further exploring the biology of LUAD, we performed functional enrichment analysis to look for biological processes and pathways related to the individual MOFA factors (see Materials and Methods section). Collectively, GSEA analysis shows an overrepresentation of MAPK and AKT on the protein level in factor 1 (see Figure 4A). The MAP kinase pathway is a highly complex signaling cascade involving three kinases. The RAS-RAF-MEK-ERK pathway is altered in forty percent of all human cancers, mainly due to mutations in BRAF and its upstream activator RAS [167]. In lung cancer, KRAS mutations often play a role in activating the MAP kinase pathway [168]. Interestingly, MAP kinase activation is also underrepresented in Factor 3 on the gene expression level (see Appendix A). AKT as part of the AKT/mTOR signaling pathway may be a downstream of the PD-L1 pathway [169]. In contrast, latent factor 2 is highly negatively enriched for gene expression pathways related to immunity (see Figure 4B). Amongst the enriched Reactome pathways are innate and adaptive immunity, interferon signaling, and notably PD-1 signaling. The PD-1/PD-L1 pathway controls the induction and maintenance of immune tolerance within the tumor microenvironment. In personalized medicine, the PD-L1 status is used as a predictor for benefit from targeted therapies or immune checkpoint blockers [170,171]. Consistently, when plotting the Reactome pathways enriched in Factor 3 CNV negative weights, it captures differences associated with the immune system, such as innate immune response, immune surveillance and inflammation (Figure 4C). For example, NFKB signaling has been demonstrated to be implicated in lung cancer manifestation, by promoting anti-tumor T cell responses [172]. Another interesting finding is that Factor 3 is also enriched in Notch related signaling pathways. It is worth noting that Notch signaling has been shown to play a pivotal role in lung cancer progression-especially in NSCLC-with genetic alterations associated with survival estimates and therapeutic significance [173,174].

As the resulting MOFA factors can be utilized to predict discrete clusters of samples, we used all the inferred factors to cluster the patients in the latent factor space, implementing collectively all information from the multi-omics layers and their differential contributions. Here, as described in the Materials and Methods section, k-means clustering resulted in three discrete groups of patients. Visualization of the three resulting clusters from the integrated analysis showed a significant overlap (using Pearson chi-squared test) with various clinicopathological parameters, such as AKAP9 gene mutational status, expression subtypes and gender (Appendix A). Of note, cluster 3 did not contain any TRU expression subtype samples, whereas the vast majority of samples harboring AKAP9 mutations were allocated in cluster 2. Finally, cluster 2 was more enriched in female patients and cluster 1 had the smallest number of mutations in the COL3A1 gene.

Finally, in order to investigate if any of the inferred latent Factors could be associated with building clinical models of predicting patient outcome, we implemented Cox proportional hazards models (coxph function from R package survival). From the identified 15 MOFA factors, Factor 1 (*p*-value = 0.0358), Factor 4 (*p*-value = 0.04162) and Factor 9 (0.04750) were statistically significantly associated with overall survival as the response variable, using *p*-values derived from Wald statistic (Figure 5).

### 2.5. Supervised Multimodal Classification Case Study with netDx

The performed supervised multi-omics data integration on CLL data using netDx resulted in a performance accuracy of 93% (±1.5%). The model was able to clearly discriminate the samples into the binary classes defined by the IGHV mutation status (AUROC = 97.1 ± 2.3%, AUPR = 92.0 ± 2.7%) (see Appendix A). Running netDx with 1 CPU took about 72 minutes for this dataset with defined settings (see Materials and Methods section). The aim of applying netDx was to obtain patient similarity networks (PSN) and group patients based on a multi-omics profile [164]. The PSN networks consist of nodes which represent the patients connected by edges representing the weighted pairwise similarities between patients. The classification of the performed analysis is based on a separation of IGHV mutation status for CLL patients which is known as a relevant prognostic factor [176]. Here we followed one suggested design of netDx developers which groups biological pathway enrichments based on gene expression measurements (Pai, et al., 2020). Selected features require a minimum feature score of 9 in at least 50% of train/test splits. The resulting pathway enrichment networks based on the expressed genes are shown for non-IGHV-mutated patients in Figure 6 and for IGHV-mutated CLL patients in Figure 7.

Each network has been manually selected as input for annotation with AutoAnnotate [177]. The titles of annotated networks in Figure 6 and Figure 7 correspond to main themes and categories within each network (e.g., mTOR signaling events) and in some cases had to be manually curated to reflect the included nodes. Enrichments of all signaling pathways from IGHV-mutated samples are included in enrichments from samples without IGHV mutations. Non-mutated samples show 13 additional pathway enrichments, which are not present in the mutated samples. The majority of shared enrichment signals have a higher score for not mutated samples (cholesterol homeostasis, mTOR signaling, phospholipases signals, interleukin 8 (IL8) and chemokine receptor 1 (CXCR1) signals and Aurora b signaling). Only the cell division control protein 42 (CDC42) signal enrichment is higher in mutated samples. The mTOR pathways are enriched in the pathway networks of both classes (IGHV- mutated and not mutated) and connected to CDC42 signals in both classes. Non-mutated samples show more concatenation of the mTOR-pathway and higher scoring to other signals. High scoring of adipogenesis and cholesterol signaling in both classes can also be observed. Ten nodes in non-mutated samples and 6 nodes in mutated samples are not connected with any edges. Figure 7 shows a relatively sparse representation of pathway networks for IGHV-mutated CLL patients in comparison to Figure 6. Two major networks are visible which mainly include IL8, CXCR1, mTOR-pathway and CDC42 signals. The enrichment called biocarta ppara in Figure 6 and Figure 7 refers to the Mechanism of Gene Regulation by Peroxisome Proliferators via PPAR alpha.

Shared enrichment between the classes show general cancer associated signals. Enrichment of adipogenesis indicates cancer-induced changes to the regulation of adipose tissue, which promotes cancer cell survival during therapy [178]. Also a breakdown of Cholesterol homeostasis is known to be linked to hypocholesterolemia in lymphocytic leukemia [179]. In addition, chemokine receptors CXCR1/2 and their ligand CXCL8 are essential for the activation and trafficking of inflammatory mediators as well as tumor progression and metastasis [180]. The IL-8 and CXCR1 related pathways are enriched in both classes. IL-8 is known for B-cell progression [181] and Chemokine receptors CXCR1/2 and their ligand CXCL8 are essential for the activation and trafficking of inflammatory mediators as well as tumor progression and metastasis [180]. Interestingly, it has been shown that leukemic B-cells neither express CXCR1 or CXCR2 nor they respond to exogenous IL-8 in CLL patients [181]. The aggregated collection of pathways highlights the mTOR –pathway, which plays a critical role in leukemia initiation [182]. Inhibitors of the mTOR-pathway are currently one line of therapies for leukemia patients. For example, CDC42 signaling as part of the mTOR-pathway and related pathways are known as key targets for CLL treatment with lenalidomide [183]. Differences in pathway related networks should highlight the driving variance of the IGHV mutation status in the gene expression layer. The Tumor Protein P63 (TP63) related pathway in non-mutated samples is completely absent in mutated samples, which demonstrates the prognostic relation between TP63- related pathways and IGHV mutation status in CLL patients [184].

Additionally to the described pathway enrichment analysis, we also performed clustering of patients based on multi-omics profiles. The following analysis is based on more strict selection of features with a minimum feature score of 9 for at least 70% of splits. Figure 8A shows the patient similarity network (PSN), which integrates the predictive features for all patient labels. Another visualization of a PSN is a tSNE plot for a cluster representation of patient classification based on the IGHV mutation status, as shown in Figure 8B. Patient similarity networks show a complex landscape of similarities where the tSNE applied clustering shows more distinct clusters of patients but not a clear separation of the IGHV mutation status. Appendix A shows how well the patients are clustered in the patient similarity network (PSN) by using pairwise patients’ shortest distance in the classes (IGHV_0 and IGHV_1) and between the classes. Distances within the classes should be smaller than between the classes in order to separate clustering of patients in the PSN.

## 3. Discussion

In the last decade, new massively parallel sequencing technologies have yielded new biological insights at the RNA, DNA, cellular, and spatial resolution, resulting in the accumulation of massive amounts of genomics data. Recently, there is a growing interest in integrating diverse data from such distinct molecular layers, in order to shed light on the biology of various complex phenomena. The simultaneous examination of multi-layer views can paint in-depth molecular pictures that provide comprehensive insights into the way our “omes” interact in the manifestation of diseases like cancer. Indeed, multimodal data integration approaches are redefining precision oncology through the exploitation of different molecular entities, which characterize holistically the molecular landscape of distinct tumors and facilitate the identification of actionable targets with clinical utility. However, despite the fact that multi-omics integration is an active area of translational cancer research, it lacks established performance benchmarks and assessment standards.

On this premise, in this review we sought to create a detailed catalogue of all the available computational tools, which a researcher could utilize for the integration of heterogeneous cancer genomics data in the context of translational cancer research. Our main goal was not only to provide a rich resource of cutting edge technologies, but also to implement two reproducible case studies, that illustrate the analysis of heterogeneous cancer multi-omics data using state of the art tools, focusing on two typical research scenarios: the supervised approach, for when a researcher for example tries to find the features that characterize the taxonomy between known molecular cancer subtypes; and the unsupervised one, that tries to unravel the heterogeneity and stratify the cancer patients into new disease subgroups. These two case study examples can serve as start-to-end workflows, which a user can utilize to analyze from scratch public multi-layered cancer data. Both studies cover important parts from multi-omics data acquisition, preprocessing of individual omics layers, integration, model training and functional enrichment analysis, along with extensive documentation of the R code and can be directly obtained from github (see here: https://github.com/Jasonmbg/CaseStudy_MAE_TCGA_LUAD_Review and https://github.com/jonasboh/Case_Study_netDx_CLL). Altogether, these two reproducible pipelines can serve as a complement to the literature mining process, on how to address two different research scenarios based on the general categorization of model nature orientation (Supervised vs. unsupervised methodologies).While it lay beyond the scope of our current review to critically analyze and provide methodological insights for each mentioned tool, we addressed the main pros and cons of the unsupervised and supervised methodologies utilized in our case studies.

This was initially demonstrated in the LUAD dataset case study, where MOFA/MOFA+ was capable of recovering known sources of biological variation related to lung adenocarcinoma expression subtypes, gender and specific somatic mutations. The molecular basis of these inferred factors aligns well with previous studies, highlighting crucial signaling pathways and perturbed biological mechanisms related to immune response, cell cycle, MAPK signaling cascades and inflammation [185,186,187]. In addition, the model identified putative clinical markers. For example, based on the top weights, using the loadings of each feature in the gene expression data, Factor 1 was highly correlated with the pulmonary-associated surfactant protein B (SFTPB) gene. A recent study illustrated that SFTPB gene expression was correlated with tumor-infiltrating lymphocytes (TIL), defining an “inflamed” lung adenocarcinoma subtype with favorable survival estimates [188]. The methodology is fast (the model was trained in less than 30 min in “medium” mode on a laptop with 64-bit Windows 10 operating system, i7 CPU 1.8GHz and 16 GB RAM), sparse and can cope with missing values. However, MOFA+ also suffers from some general limitations of unsupervised data-integration methodologies. In detail, while matrix factorization techniques are often used to reduce the feature space from tens of thousands to a significantly lower number, they might inadvertently ignore a large amount of biological information concerning relationships between features. The same caveat is intrinsic to MOFA+, as the model assumes independence between features in its prior distribution. Furthermore, MOFA+ by nature is limited to capture strong non-linear relationships, which could be an issue when trying to analyze noisy datasets with high non-linearities as these would result in small amounts of variance explained. Overall, while matrix decomposition methodologies are quite popular as the method of choice among the available unsupervised data integration approaches, the biological interpretation of the inferred latent factors can be a challenging process [189]. A representative example is somatic mutations: latent factors are essentially defined as linear combinations of features. Thus, for a factor to exist it requires an effect over multiple features. Sometimes, somatic mutations don’t “behave” like this, as a single somatic mutation can produce a large downstream effect on the expression level, rather than having a contribution to a single factor. This was also evident in our analysis, where based on an initial exploratory training of the model, the plot of ‘resulted variance explained’ showed that the somatic mutations did not have a contribution over the factors, and thus seemed to behave as independent features.

The performed supervised analysis for patient classification with netDx based on patients suffering from chronic lymphocytic leukemia (CLL) illustrates a typical use case for supervised multi-omics data integration. The challenge here lies in applying multi-omics data integration on a sparse cohort of less than 1000 patients, with missing values, unequal feature sizes per layer and unequal class sizes (class imbalance). The used cohort includes clinical, methylation, drug response, and gene expression data. Unfortunately, we were not able to integrate mutation data, likely because of the sparsity of this data layer which comprised of the binary mutation status for 69 genes. The aim of this case study was to show the challenges and possibilities when working with multi-omics data for a specific research purpose. Supervised analysis in the multi-omics field is based on prior knowledge of the data and its biomedical context, which impacts both hypothesis and primary feature selection. In our study we classified CLL patients based on their IGHV mutation status in order to separate patients with better treatment response prognosis (IGHV-mutated) from those with a worse prognosis (no IGHV mutation). The majority of findings in the netDx study is based on the gene expression layer, which was used for pathway enrichment analysis (see Figure 5 and Figure 6).

The classification of CLL patients based on IGHV status resulted in multiple clusters (see Figure 8B). The implementation of pathway enrichment analysis for methylation and drug response layers is likely to have a great effect on the separation of classes in resulting PSN. The interpretation of results also needs to take into account the differences in class imbalance.

The limitations of this study are not only based on the aforementioned data related issues but also due to the nature of supervised methodologies. Supervised approaches need to be validated on an external dataset in order to evaluate potential overfitting and the generalizability of predictions. Therefore, the good performance of netDx needs to be validated on an independent larger dataset using the same features as in the applied model. Observed differences in performance between netDx v. 1.2.2 and 1.3.1 led us to apply netDx v1.3.1 as an application under development in a Docker container. Increasing the number of performed cross validations could identify more clearly generalizable patterns in a small or very heterogeneous dataset, but it would further increase the calculation time. Fortunately, there are more enriched pathways for non IGHV mutated patients, which have a worse prognosis, than for IGHV mutated patients. Targeted therapy of these patients based on their enriched pathways could lead to better prognosis for these patients. In summary, we could identify enriched pathways which are known to be involved in the pathophysiology of CLL. Furthermore, we could highlight interconnected pathways in the mTOR and the TP63 network in non-IGHV mutated samples. After validation of these results, they could help lead to the refinement of existing treatment combinations for targeting the aforementioned enriched networks for IGHV-mutated and non-IGHV mutated patients.

Furthermore, it is essential to highlight some putative limitations of our literature review. The literature research has been performed in an automated framework by using the Entrez Direct (EDirect) tool. General limitations of automated annotations concern the challenge to classify literature without evaluating its context. For example, publications of tools such as SIMMS [190] failed to be considered in our literature search, as they use the word multi-modal which is not used in our mining process as it is more general and increases the number of unspecific search results. Although the classification of papers in our review is partly based on occurrence of words in title and abstract, which does not necessarily correspond to the content of the paper, we can see from the results shown in Appendix A and categorization of multi-omics integration tools in Table 2, Table 3, Table 4, Table 5, Table 6 and Table 7, that the classification of mined literature into five categories worked quite well. The selected papers for tool classification all refer to the corresponding general research purpose. Another limitation of this research is the sole use of PubMed as literature database, which may exclude some papers such as more technical oriented literature, However, PubMed contains cancer-related publications and thus suited our motivation of providing an overview of literature for cancer-related multi-omics data integration.

Finally, in addition to the main aforementioned challenges that govern the integration, sharing and utilization of distinct omics sources, we would like to summarize three major aspects, which facilitate the robust and successful amalgamation of heterogeneous cancer data layers:Initially, one important aspect that influences the integration part includes the pre-processing steps prior to integrating any multimodal data: initially, appropriate normalization or transformation is essential to remove any technical confounders related to each omic data layer. For example, for count based data like RNA-Seq, size factor normalization and variance stabilization are generally recommended. Also, significant differences in size in at least one of the interrogated omics layers could inflate the data integration model to capture non-biological variation associated with this specific data layer, while downweighting more subtle sources of variation. In addition, it is well known that most genomic studies suffer from the “curse of dimensionality”, that is the number of features being substantially higher than the number of samples. Hence, a feature selection step like selecting the top most variable features per omic modality is essential, both in supervised and unsupervised approaches. However, filtering is not trivial especially when dealing with somatic mutations or copy number alterations, where a more “sophisticated” filtering is needed. Somatic mutations can be very sparse with the vast majority of cancer genes being of low prevalence, cancer-specific and not shared among all patients of the same cancer. Intratumoral diversification adds further complexity to the application of a simple reduction based on the frequency of no events [191]. Instead, clinical data portals with prior biological knowledge should be used along with computational frameworks to identify putative driver genes, aiming to reduce the CNV/somatic mutations feature space. A quality control step is critical to investigate the percentage and distribution of missing values relative to the number of total samples. In the near future, improvements of the human reference genome (GRCh38) could increase completeness of multi-omics studies. For example, applied telomere to telomere long-read sequencing has started to fill unresolved gaps in the human reference genome for the X Chromosome [192]. Further ongoing efforts will reveal new functional landscapes by creating a human pan-genome, which would include diverse sets of individuals in order to catch the genomic variation across different populations [193]. This will provide the opportunity to study genetic similarities and differences among human populations within genomic or multi-omics studies of complex diseases.Additionally, another crucial part lies in the biological interpretation of the integrative analysis: it is vital to associate any findings to molecular mechanisms and perturbed pathways, in order to identify any causal regulatory relationships between the profiled entities. For this purpose there are various recent tools and databases that perform pathway analysis and provide prior knowledge on molecular biology to construct intracellular communication networks well-suited for multi-omics functional annotation. These include but are not limited to SignaLink 2.0 [123], OmniPath [124,194], ReactomeGSA [195] and ActivePathways [196]. Moreover, incorporation of prior knowledge from clinical data portals could further facilitate the prioritization of features or signatures from multimodal studies, which could serve as putative biomarkers. A representative example is the Variant Interpretation for Cancer Consortium (VICC) meta-knowledgebase [197], a harmonized effort for cancer variant interpretation by encapsulating multiple different cancer variant annotation databases. VICC can be utilized for the validation of putative biomarkers from multi-omics cancer studies. Consequently, the development of multi-omics data integration methodologies that incorporate such prior biological knowledge should be enhanced as well, in order to delineate more readable causal networks between the perturbed omics in cancer manifestation. COSMOS (Causal Oriented Search of Multi-Omics Space) for example integrates phosphoproteomics, transcriptomics, and metabolomics data sets with prior knowledge such as protein-protein interactions to create hypotheses about causal links between signaling kinase cascades, transcriptional factors and metabolites [198].Finally, a researcher should strongly consider to follow specific protocols such as the FAIR guiding principles (findability, accessibility, interoperability, and reusability) [199] when publishing multi-omics cancer data, and to take into account important bioethics considerations when sharing cancer patient data [200].

## 4. Materials and Methods

### 4.1. Literature Review Workflow

A systematic and automated literature search in the PubMed database was performed with the aim to collect publications of interest and classify them into distinct meaningful classes without manual configuration. Based on fast rising numbers of publications in the field of multi-omics data integration, the here portrayed results may change heavily in the future. For literature mining we employed the Entrez Direct (EDirect) tool, which allows systematic filtering of publications when accessing the NCBI publication databases, and query for multiple molecular data types in a command-line frame [201]. EDirect facilitates a multi-step search in one single command with the use of piped command blocks. The used keywords mesh terms, and additional search criteria were chosen to be specific for multi-omics data integration in the field of multi-view learning (see Table 8). We refrained from the use of multi-view learning as a search term itself, as multi-view data refers to the general use of any kind of heterogeneous data [202] and associated techniques are also widely applied for non-clinical purposes [203]. Multi-view applications should therefore be applied carefully in the multi-omics context in the literature search [204].

The retrieved publications are listed in Appendix A and include the publication date, the PubMed ID and the title. Furthermore, we defined five different classes with specific Entrez search filtering criteria in order to define the purpose of the paper (cancer, review, or tool) and the nature of the presented methodology (supervised or unsupervised). In order to keep it simple and clear, we did not use more complex classifications like semi-supervised or recurrent learning. For a detailed description of EDirect commands see the Appendix A section.

### 4.2. LUAD Dataset and MOFA Analysis

Despite the significant advances in targeted treatments with receptor tyrosine kinase inhibitors like Sunitinib or immune checkpoint inhibitors, lung cancer remains the first leading cause of cancer-related deaths and the second most commonly diagnosed cancer worldwide in both sexes, based on the WHO GLOBOCAN database (http://gco.iarc.fr/today/fact-sheets-cancers) epidemiological data for 2020. This can be largely attributed to its propensity to metastasize to the brain, and its high lineage plasticity resulting in poor prognosis and treatment relapse [171,205,206]. Amongst the two major types of lung cancer, namely non-small cell lung cancer (NSCLC) and small cell lung cancer (SCLC), lung adenocarcinoma (LUAD) is the most common histological subtype. Overall, lung cancer is considered a highly heterogeneous disease, with complex etiology.

To use a reproducible case study example for the integration of heterogeneous public cancer multi-omics data with an unsupervised approach, we retrieved and downloaded the LUAD TCGA cohort genomic cancer multimodal dataset [168] using the R package curatedTCGAData (v. 1.12.0). The complete bioinformatics analysis was performed with custom made scripts in R-4.0.3/Bioconductor. Briefly, the four omics layers gene expression, proteomics, copy number variation, and somatic mutations were selected for an initial number of patients, utilizing the MultiAssayExperiment integrative data container R package (v. 1.16.0). The total number of patients and features for each assay are illustrated in the Appendix A. Using the R package TCGAutils (v. 1.10.0) for initial pre-processing, the final number of common patients profiled across all the assays was 181. For the gene expression data (Upper quartile normalized RSEM TPM gene expression values) downstream analysis included normalization using the variance stabilizing transformation (VST) from DESeq2 (v1.30.0) [207]. Then, we applied a non-specific intensity filtering to remove genes that are not expressed in more than 50% of all samples. Afterwards, additional variance filtering for dimensionality reduction was performed using M3C (v. 1.12.0) [208], resulting in 6958 genes remaining in the dataset.

For an extended feature reduction, we selected only those genes from the copy number alteration and the somatic mutation data that overlapped with the aforementioned final expression genes. In addition, for the somatic mutations, we further performed an intersection of the top 100 most frequently mutated genes, with the COSMIC, Cancer Gene Census (CGC) gene list [209]. This resulted in 13 common genes, which were used as external clinical covariates for downstream analysis. From the protein data we only removed those proteins that had missing (NA) values in the majority of samples, as the downloaded RPPA data were already normalized.

After data preprocessing we used the R package MOFA+ (1.0.1) [155], an unsupervised factor analysis model to perform multi-omics data integration based on the three layers expression, copy number alterations and proteins, while the somatic mutations were used as external clinical covariates. For model training we used default parameters (number of factors = 15, convergence mode = “medium”). To investigate and interpret the output of the model, we utilized various internal package functions. Additionally, we applied principal component gene set enrichment (PCGSE) [210] with Reactome gene sets [211] downloaded from MSigDB [121,212] to interrelate the inferred latent factors to biological processes and molecular pathways. Finally, we isolated all the numeric inferred latent factors to conduct unsupervised clustering of the patients in a multi-omics fashion, with the ultimate goal of predicting discrete clusters that could resemble disease subtypes. For the selection of the optimal number of clusters we utilized the M3C package [208] (Monte Carlo iterations = 100, resampling reps for reference-real data = 250, inner clustering algorithm = kmeans). Visualization of the resulting clusters along with available clinicopathological data was conducted using the ComplexHeatmap R package (v. 2.6.2) [213].

### 4.3. CLL Dataset and netDx Analysis

Chronic lymphocytic leukemia (CLL) is the most common type of adult leukemia in the western world. CLL is known as a chronic disease affecting B lymphocyte activity. B cells are activated by different stimuli of the B cell receptors (BCR) coming from cytogenetic abnormalities as well as different genetic alterations. Most of CLL patients carry at least one of four common chromosomal alterations, namely deletion 13q14, deletion 11q22-23, deletion 17p12, and trisomy 12. Frequently, mutations include genes that can be integrated into the NOTCH signaling, inflammatory receptor, MAPK, NFKB, DNA damage and cell cycle control, chromatin modification, transcription, and ribosomal processing pathways [214]. However, the underlying role of included genetic alterations for development and progression of CLL is still largely unknown. There is also growing evidence which implicates aberrant signaling through the mTOR pathway in B cell malignancies [182,215,216].

A multi-omics drug perturbation study [217] using CNV, methylation, mutation, gene expression, and drug response measurements clustered 246 CLL patients into three groups based on their drug response. These groups were separated by signals belonging to the BCR pathway, the mTOR pathway or MEK pathway. The study highlights the IGHV gene mutation status and trisomy 12 as very important markers of kinase inhibition in their integrated analysis.

Somatic mutations of the IGHV gene are known to be prognostic clinical markers for chemoimmunotherapy outcome and therefore crucial factors for patient survival [176]. Analysis of a subcohort using the MOFA algorithm reported the somatic mutation status of the IGHV and trisomy 12 as driving sources of molecular heterogeneity of CLL [155] The subcohort comprised 200 patients with CLL including gene expression data (5000 features, 136 samples), mutation data (69 features, 200 samples), methylation data (4248 features, 196 samples), and drug response data (310 features, 184 samples).

In the analysis the IGHV status was linked to the differentiation of cancer cells and the activation of B-cell receptors, and is the main factor driving the variance in the gene expression layer of the used CLL cohort.

We chose this subcohort for supervised multi-omics classification on the IGHV mutation status using netDx [107]. One motivation was to complement an unsupervised analysis with a supervised one to increase the evidence for the importance of the IGHV status in CLL. Another motivation for selecting this dataset was that it represents well the challenges for multi-omics integration tools with few samples (<1000), unequal distributed missingness (136–200 samples), and unequal feature representation for different layers (69–5000 features per layer). The analysis was performed with R/Rstudio.

We applied netDx v. 1.3.1 with R-4.0.3 by using one CPU on a MacBook Pro with macOS Big Sur 11.1, 3.1 GHz Quad-Core Intel Core i7 and 16 GB memory. We used a Docker container with preinstalled R and netDx as well as all dependencies (https://hub.docker.com/repository/docker/shraddhapai/netdx), using the gene expression, the drug response, and the methylation data layer. The input was defined by the binary IGHV mutation status (0 = not mutated, 1 = mutated). Having to remove 28 samples based on the missing mutation status of the IGHV gene, the final input dataset contained 172 samples (98 with and 74 without IGHV mutation). No further data filtering based on missingness was applied, as netDx can handle missing values in different omics layers. The design of the features for patient classification was grouped by pathways for gene expression data and one feature per layer for the others. We compiled 728 pathways containing 10–200 genes from several curated pathway databases [107]. We used 10 train/test splits with 80% of every split for training and 20% for testing. Feature selection was performed by setting the maximum feature score to 10 (featScoreMax) and the feature selection threshold to 9 (featSelCutoff). Only features with minimum netDx scoring of 9 were further used for classification of patients in the test set. Well-performing features were selected based on performance across train/test splits. Features needed to score at least 9 in at least 50% of splits. Selected features for the creation of final patient similarity networks (PSNs) need to pass a more strict selection of a minimum score of 9 in at least 70% of splits. For visualization of enrichment maps and similarity networks we applied Cytoscape v. 3.8.2 with EnrichmentMap app v. 3.3.1 and an edge cutoff on similarity of 0.1. See corresponding GitHub repository for detailed step by step procedure.

## 5. Conclusions

For one, we conclude that automated literature search, although not a guarantee for accurate or comprehensive results, gives a good overview and classification of published knowledge in a specific field, in this case the integration of multi-omics data in oncology. Secondly, the findings of our case studies demonstrate that we can retrieve both known results and novel findings using predominantly the core tools with minimal tuning. Furthermore, in the future the definition and improvement of data sharing and biomedical meta-data to enhance clinical decision support will be of critical importance. Finally, bringing together interdisciplinary computational teams and researchers will help promote the development of cutting edge techniques for multi-omics integration and analysis, increasing the necessity for multi-platform analysis with common datasets to benchmark the performance of various methodologies.

## Figures and Tables

**Figure 1 ijms-22-02822-f001:**
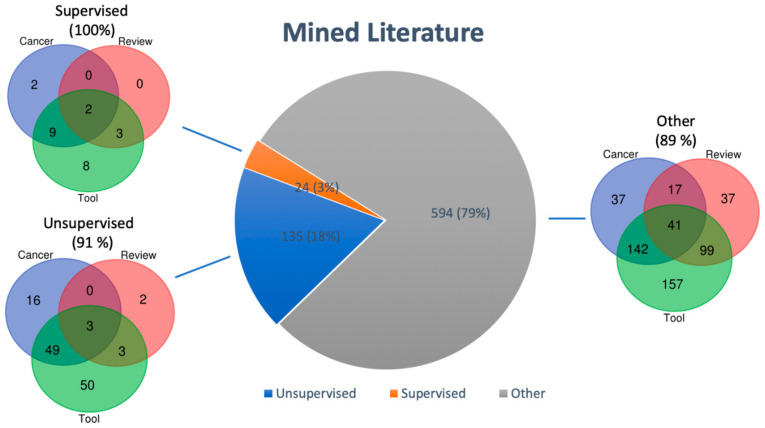
Summary of literature mining. The general search can be classified into three main classes Supervised, Unsupervised and Other. The majority of papers (100% in Supervised, 91% in Unsupervised, 89% in Other) within these classes are included in the overlapping subclasses Cancer, Review, and Tool (Venn diagrams have been created with http://bioinformatics.psb.ugent.be/webtools/Venn/).

**Figure 2 ijms-22-02822-f002:**
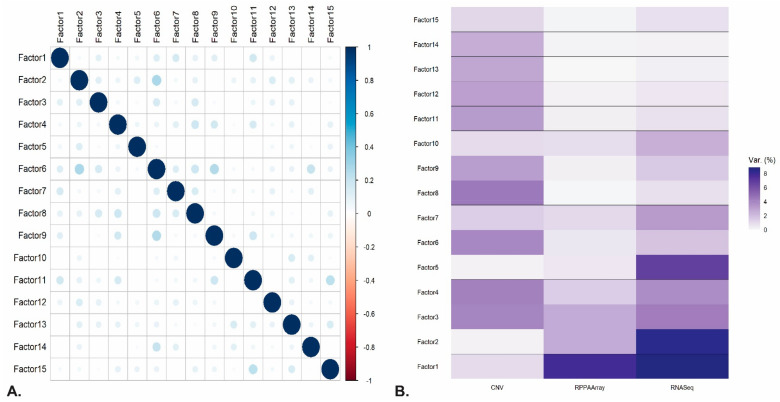
(**A**). Plot of the correlation matrix between the inferred Multi-Omics Factor Analysis (MOFA) latent factors, which can be used as a quality control of the fitted model. It returns a symmetric matrix with the correlation coefficient between every pair of factors. Blue color denotes positive correlation, whereas red negative, respectively. A diagonal correlation matrix is usually expected for a robust model fit, suggesting low correlation overall between the MOFA factors. (**B**). Variance decomposition analysis plot, which illustrates the variance explained (R-squared value) per factor and per layer (CNV, RPPAArray = protein, RNASeq = expression). The values are calculated using a coefficient of determination, which ranges from 0 to 1, and scaled to a percentage by multiplying by 100.

**Figure 3 ijms-22-02822-f003:**
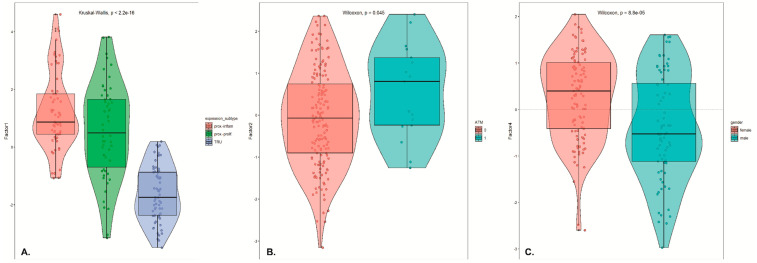
(**A**). Visualization of samples using Factor 1 values, which are colored by the covariate expression_subtype, which denotes the 3 available lung adenocarcinoma transcriptional subtypes. The plot shows a clear separation of the “TRU” subtype from the “proximal-proliferative” and “proximal-inflammatory”. (**B**) Visualization of the samples using Factor 2 and ATM mutation status for color. Samples with positive values have the ATM mutation *blue), whereas samples with negative Factor 2 values do not have the mutation (red). (**C**) Visualization of the LUAD samples using Factor 4 and the gender covariate to color the selected factor values. From the relative plot the significant association of Factor 4 with gender is illustrated. Samples with average positive values are mostly female and samples with negative values are mostly male. In all plots, *p*-values for comparison of the means of the groups were calculated using the function stat_compare_means from ggpubr R package.

**Figure 4 ijms-22-02822-f004:**
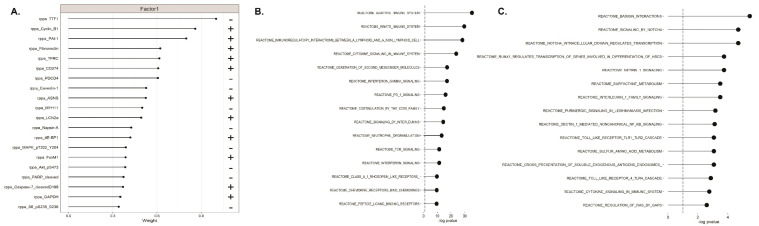
(**A**) Line plot displaying the absolute loading from the top 20 features of Factor 1 in the protein data. The corresponding weight sign is depicted on the right, scaled from −1 to 1. Proteins with positive weights have higher levels of expression in the samples that have Factor 1 positive values, and vice-versa. (**B**) Visualization of the enrichment analysis results, running GSEA on Multi-Omics Factor Analysis (MOFA) factor 2 with gene expression negative weights and Reactome gene sets. (**C**) Visualization of the enrichment analysis results, running GSEA on MOFA factor 3 with copy number variation negative weights and Reactome gene sets.

**Figure 5 ijms-22-02822-f005:**
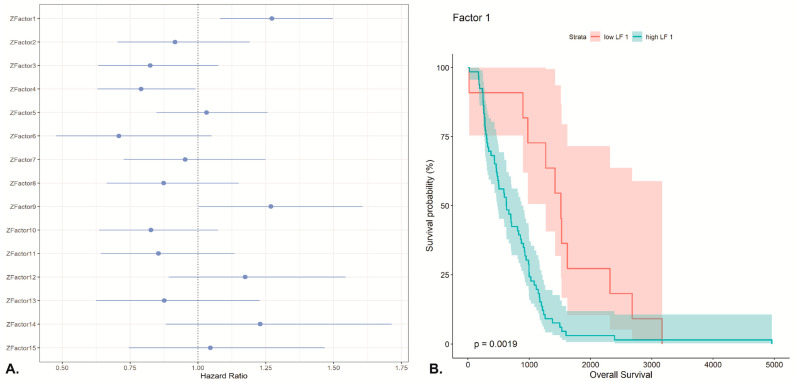
Putative prognostic utility of the Multi-Omics Factor Analysis (MOFA) latent factors. (**A**). Forest plot of resulting hazard ratios, illustrating the association of the tested MOFA factors with overall survival, based on Cox regression modeling (error bars representing 95% confidence intervals) (**B**) Example of a Kaplan-Meier plot for Factor 1, showing overall survival. The samples were separated into two distinct groups, based on the maximally selected rank statistics from the maxstat R package [175]. As Factor 1 has a positive coefficient, samples with high values have an increased hazard in comparison to samples with low relative values. The *p*-value was calculated using a log-rank test on the two aforementioned groups.

**Figure 6 ijms-22-02822-f006:**
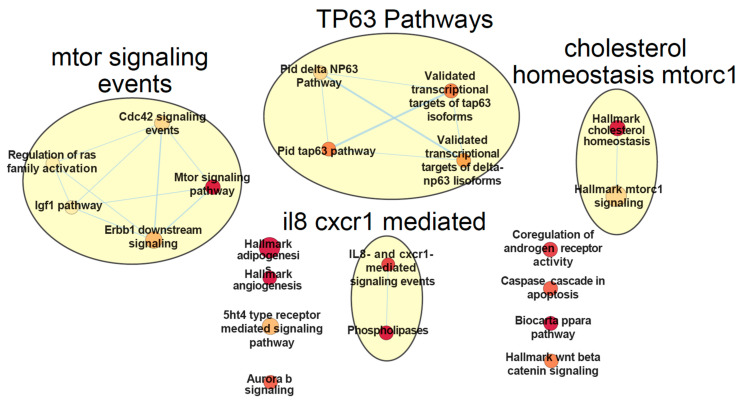
Top predictive features for not mutated IGHV CLL patients. Each node shows predictive pathway features and edges, which connect shared genes of pathways. Node fill uniformly indicates highest score; yellow = netDx score 3, red = netDx score 10. The size of the nodes displays the amount of genes in the underlying gene set. Selected features required a minimum feature score of 9 out of 10 in at least in 50% of train/test splits.

**Figure 7 ijms-22-02822-f007:**
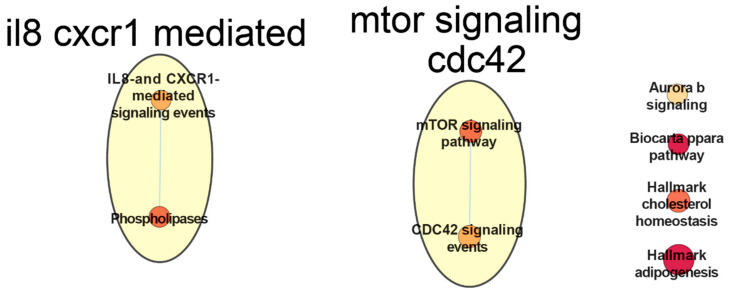
Top predictive features for IGHV-mutated CLL patients. Each node shows a predictive pathway features and edges connect shared members of pathways. Node fill uniformly indicates highest score; yellow = netDx score 3, red = netDx score 10. Selected features required a minimum feature score of 9 out of 10 in at least in 50% of train/test splits.

**Figure 8 ijms-22-02822-f008:**
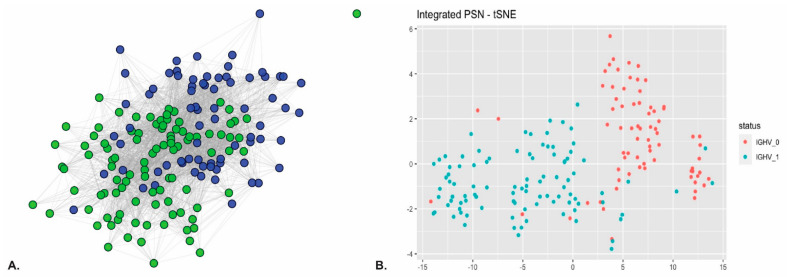
(**A**). Integrated patient similarity network for binary stratification of IGHV status based on multi-omic data (green—IGHV mutation, blue—no IGHV mutation). Each node in this network corresponds to a CLL patient and each edge corresponds to weights displaying the average similarity across all features passing feature selection [164]. This network was generated by implemented functions of netDx and visualized by Cytoscape. (**B**) tSNE visualization of integrated patient similarity network for binary stratification of IGHV status (IGHV_0 = no IGHV mutations, IGHV_1 = IGHV mutations). Only features which passed the feature selection step are integrated.

**Table 1 ijms-22-02822-t001:** Different omics levels of gene-function relationship.

Level of Analysis	Definition	Method of Analysis
Genome [4]	Complete set of genes of an organism or its organelles	WGS, WES, DNA microarray
Transcriptome [5]	Complete set of messenger RNA molecules present in a cell, tissue of organ	RNA-Sequencing Expression microarray Spatially resolved transcriptomics
Proteome [6]	Complete set of protein molecules present in a cell, tissue or organ	Peptide/protein microarrays (RPPA) Mass spectrometry Imaging mass cytometry
Metabolome [7]	Complete set of metabolites (low-molecular-weight intermediates) in a cell, tissue or organ	Nuclear magnetic resonance spectrometry Mass spectrometry Infa-red spectroscopy
Methylome [8]	Complete set of methylation sites within a genome	Bisulfite-Sequencing, ChIP-Seq
Microbiome [9]	Complete set of genes of all microbes (bacteria, fungi, protozoa and viruses) in a cell, tissue or organ	DNA-Sequencing 16S rRNA-Sequencing
Lipidome [10]	Complete set of all biomolecules defined as lipids	Mass Spectrometry

WGS: Whole-genome Sequencing; WES: Wole-exome sequencing; ChIP: Chromatin Immunoprecipitation.

**Table 2 ijms-22-02822-t002:** Summary of computational tools for patient stratification in the field of multi-omics data integration.

Model Nature Orientation	Tool Name	Programming Language	Integration Method	Used Omics	Reference
Unsupervised	R.JIVE	R	Multi-step analysis	mRNA, miRNA, MET	[104]
Unsupervised	PROFILE	R, Python	Multi-step analysis	mRNA, CNV, MUT	[105]
Supervised	SALMON	Python	Gene co-expression Analysis	mRNA, miRNA, CNV	[106]
Supervised	netDx	R	Feature network aggregation	mRNA, miRNA, CNV, MUT, MET, PROT	[107]
Supervised	Graper	R	Bayesian	mRNA, DRUGre, MET	[108]

mRNA = Gene Expression, miRNA = microRNA Expression, MET = Methylation, CNV = Copy Number Variation, PROT = Proteomics, DRUGre = Drug response, MUT = mutation.

**Table 3 ijms-22-02822-t003:** Summary of computational tools for biomarker discovery in the field of multi-omics data integration.

Model Nature Orientation	Tool Name	Programming Language	Integration Method	Used Omics	Reference
Unsupervised	Joint Bayes Factor	Matlab	Matrix factorization	mRNA, MET, CNV	[111]
Unsupervised	iProFun	R	Multiple-step analysis	mRNA, CNV, MET	[112]
Unsupervised	CCA-sparse group	Matlab	Canonical correlation analysis	mRNA, SNP	[113]
Supervised	sMBPLS	Matlab	Partial Least Squares	mRNA, miRNA, CNV, MET	[114]
Unsupervised	CNAmet	R	Multi-step analysis	mRNA, CNV, MET	[115]
Supervised	iBAG	R	Multi-step analysis	mRNA, CNV, MET	[116]
Supervised	Anduril	R, Python, Shell	Multi-step analysis	aCGH, mRNA, miRNA, SNP, MET	[117]
Supervised	CapsNetMMD	Python	Capsule network model	mRNA, CNV, MET	[118]

mRNA = Gene Expression, miRNA = microRNA Expression, MET = Methylation, CNV = Copy Number Variation, aCGH: DNA microarray.

**Table 4 ijms-22-02822-t004:** Summary of computational tools for pathway analysis in the field of multi-omics data integration.

Model Nature Orientation	Tool Name	Programming Language	Integration Method	Used Omics	Reference
Unsupervised	ModMap	Java	Multi-step analysis	mRNA, miRNA, PROT	[125]
Unsupervised	NetICS	Matlab	Multi-step analysis	mRNA, miRNA, CNV, MUT, MET, PROT	[126]
Unsupervised	SNMNMF	Matlab	Matrix factorization	mRNA, miRNA	[127]
Unsupervised	PARADIGM	Web-app, Python	Probabilistic graphical models	mRNA, CNV	[128]
Supervised	FSMKL	Matlab	Multiple kernel learning	mRNA, CNV	[129]
Unsupervised	Sumer	R	Multi-step analysis	mRNA, PROT	[130]
Unsupervised	MOSClip	R	PCA	mRNA, CNV, MUT, MET	[131]
Supervised and Unsupervised	COCOA	R	Multi-step analysis	mRNA, ATAC-Seq, DRUGre, MUT, MET	[132]

mRNA = Gene Expression, miRNA = microRNA Expression, MET = Methylation, CNV = Copy Number Variation, PROT = Proteomics, DRUGre = Drug response, MUT = mutation, ATAC-Seq = Transposase-Accessible Chromatin.

**Table 5 ijms-22-02822-t005:** Summary of computational tools for drug analysis (drug repurposing and drug discovery) in the field of multi-omics data integration.

Model Nature Orientation	Tool Name	Programming Language	Integration Method	Used Omics	Reference
Supervised	MOLI	Python	Neural networks	mRNA, CNV, MUT	[134]
Unsupervised	SNPLS	Matlab	Partial least squares	mRNA, DRUGre	[135]

mRNA = Gene Expression, CNV = Copy Number Variation, DRUGre = Drug response, MUT = mutation.

**Table 7 ijms-22-02822-t007:** Summary of computational methods which can be applied to several mentioned research aims in a multi-omics context.

Research Purpose	Model Nature Orientation	Tool Name	Programming Language	Integration Method	Used Omics	Reference
Biomarker discovery, Cancer subtype classification, Pathway analysis	Unsupervised	MCIA	R	Multi-step analysis	mRNA, PROT	[146]
Patient stratification, Cancer subtype analysis	Supervised	mixOmics	R	Feature transformation	mRNA, miRNA, PROT	[147]
Biomarker prediction, Pathway analysis	Unsupervised	Lemon-Tree	Java	Module network learning	mRNA, CNV	[148]
Patient stratification, Cancer subtype classification	Unsupervised	Clusternomics	R	Multi-step analysis	mRNA, miRNA, MET, PROT	[149]
Biomarker discovery, Pathway analysis	Unsupervised	AMARETTO	R	Multi-step analysis	mRNA, CNV, MET	[150]
Pathway analysis, Cancer subtype classification	Supervised	iOmicsPASS	C++	Multi-step analysis	mRNAs, CNV, PROT	[151]
Biomarker discovery, Cancer subtype classification	Unsupervised	MOGSA	R	Matrix factorization	mRNA, CNV, Phosp, PROT	[152]
Patient stratification, Pathway analysis	Unsupervised	PathME	R, Python	Matrix factorization	mRNA, miRNA, CNV, MET	[153]
Drug analysis, Pathway analysis	Supervised	DrugComboExplorer	Java, Python	Multi-step analysis	DNA, mRNA, CNV, MET	[154]
Biomarker discovery and Patient stratification	Unsupervised	MOFA	R	Matrix Factorization	mRNA, MUT, MET, DRUGre	[155]

mRNA = Gene Expression, miRNA = microRNA Expression, MET = Methylation, CNV = Copy Number Variation, PROT = Proteomics, DRUGre = Drug response, MUT = mutation, Phosp = Phosphorylation profiles.

**Table 8 ijms-22-02822-t008:** Summary of search strategies and associated keywords for collecting multi-omics data integration associated publications with EDirect (PTYP: publication type; MESH: medical subject headings).

Search Round	Search Terms
General Search	(multi AND omics) OR multi-omics OR multiomics OR (multivariate AND genomic) OR (Algorithms AND integrative AND Cluster AND Analysis)) AND data AND integration
Searching for supervised methods	General Search + supervised
Searching for unsupervised methods	General Search + unsupervised OR cluster OR (Factor AND Analysis) NOT supervised
Searching for reviews	General Search + review [PTYP] OR review
Searching for tools	General Search + Tool OR Application OR Algorithm OR method
Searching for cancer	General Search + humans [MESH] AND cancer [MESH]) OR cancer

## Data Availability

The performed Literature search is displayed in the GitHub repository: https://github.com/jonasboh/multi-omics_literature_search.

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
