# Peer review of "A Detailed Catalogue of Multi-Omics Methodologies for Identification of Putative Biomarkers and Causal Molecular Networks in Translational Cancer Research"

_ijms, 2021, doi:10.3390/ijms22062822_

Round 1

Reviewer 1 Report

The authors made an effort in cataloging and categorizing available computational tools for multi-omics data integration
through automated online literature mining. Authors must address the following comments: 

Major Comments

1) The title of the manuscript is "Evaluation of multi-omics ...". However the authors don't really evaluate anything in the paper.As the authors point out in the discussion they "sought to create a comprehensive catalogue of all available computational frameworks". The title should be changed in something like "Cataloging multi-omics ..." or similar alternatives.   

2) Line 43 "Table 1 shows the flow of information". Table 1 doesn't show any flow. Additionally, Table 1 should be extended to include "lipidomics" and "ionomics" (see for example Figure 1 in Haas, Robert, et al. "Designing and interpreting ‘multi-omic’experiments that may change our understanding of biology." Current Opinion in Systems Biology 6 (2017)). References for Ionomics should include Baxter, Ivan. "Ionomics: studying the social network of mineral nutrients." Current opinion in plant biology 12.3 (2009); Konz, Tobias, et al. "ICP-MS/MS-based ionomics: a validated methodology to investigate the biological variability of the human ionome." Journal of proteome research 16.5 (2017): 2080-2090.

3) Section 1.4: I suggest the authors to conclude the section by mentioning online curated platforms that also provide tools to perform standardised multi-omics data analysis, for example Galaxy (https://galaxyproject.eu/citations).

4) Section 2.1: because the literature mining approach represent most of the novelty of the paper this section needs to be strengthen. I invite the authors to i) extend the text by adding also here a description of the mining procedure in Supplementary Methods, and ii) to add a multiple panel Figure to descrive the statistic of their mining. This main text figure should include Figure S2 and also the second order statistic describing co-occurrence of the selected keywords.   

5) Section 2.2.2: expand description. The enumerated tools integrate single mic data to extract multi-omics markers? Where is the multi-comic aspect of these tools ?

6) Section 2.2.4: Drug analysis, again I invite the authors to expand the text and highlight the multi-comic aspects of the tools within this analysis category.

7) Figure 1: Quality too low. Authors must provide high quality figures. Additionally they have to better describe what they show in the caption (MOFA is mentioned at the third line of caption!) 

8) Figure 2: Quality too low. Unreadable. Authors must provide high quality figures. 

9) Figure 3: Quality too low. Unreadable. Authors must provide high quality figures. 

10) Figure 4: Quality too low. Unreadable. Authors must provide high quality figures. 

11) Figure 4: Quality too low. Authors must provide high quality figures.  

Other comments 

12) Line 67 : "the vast ... as an iceberg"; reformulate the sentence. It doesn't sound good.

13) Line 70 : perception -> understanding ? What do the authors mean by "reflecting the perturbed components restricted to relative molecular layer?

14) Line 92: worth citing here also Orlandi, Ester, et al. "Potential role of microbiome in oncogenesis, outcome prediction and therapeutic targeting for head and neck cancer." Oral oncology 99 (2019), since the authors included microbiome in Table 1. 

15) Line 243: in addition to reference 43 I strongly encourage to cite here i) Haas, Robert, et al. "Designing and interpreting ‘multi-omic’experiments that may change our understanding of biology." Current Opinion in Systems Biology 6 (2017), and ii) Iacovacci, Jacopo, et al. "Extraction and Integration of Genetic Networks from Short-Profile Omic Data Sets." Metabolites 10.11 (2020).

16) Lines 248-249 : "standardized" protocol; the need for standardised analysis is an important point, please explain the concept of standardisation to facilitate readability. "Under-utilization" -> underutilization, is one word without quotes.

17) Line 305: in addition to reference 77 and 78 worth citing here the following review : Simidjievski, Nikola, et al. "Variational autoencoders for cancer data integration: design principles and computational practice." Frontiers in genetics 10 (2019).

18) Line 402-403 : dicovery -> discovery; marker -> markers.

19) Table 7: caption unclear.

20) Line 614: the main novelty of our study lies in the implementation of two reproducible cases. Meaning unclear, explain better. 

21) Line 623: "we can robustly summarize and highlight ... data integration". I don't understand this phrase.

22) Line 707: "General limitations ...'. I would add a couple of lines describing a concrete example. My suggestion: the SIMMS (Subnetwork Integration for Multi-Modal Signatures, Haider, Syed, et al. "Pathway-based subnetworks enable cross-disease biomarker discovery." Nature communications 9.1 (2018)), is a well known multi-omics pathway analysis tool in cancer research, but because it is describe by the word "multi-modal" and not "multi-omics" it wasn't detected.       

23) Line 721 and 726: why bullets points 6. And 7. ?

Author Response

Major Comments

1) The title of the manuscript is "Evaluation of multi-omics ...". However the authors don't really evaluate anything in the paper. As the authors point out in the discussion they "sought to create a comprehensive catalogue of all available computational frameworks". The title should be changed in something like "Cataloging multi-omics ..." or similar alternatives.   

We thank the reviewer for pointing this out, and we agree. Therefore, we changed the title to reflect the reviewer’s concerns to ‘A detailed catalogue of multi-omics integration methodologies for …’

2) Line 43 "Table 1 shows the flow of information". Table 1 doesn't show any flow. 

We agree and changed this sentence accordingly.

Additionally, Table 1 should be extended to include "lipidomics" and "ionomics" (see for example Figure 1 in Haas, Robert, et al. "Designing and interpreting ‘multi-omic’experiments that may change our understanding of biology." Current Opinion in Systems Biology 6 (2017)). References for Ionomics should include Baxter, Ivan. "Ionomics: studying the social network of mineral nutrients." Current opinion in plant biology 12.3 (2009); Konz, Tobias, et al. "ICP-MS/MS-based ionomics: a validated methodology to investigate the biological variability of the human ionome." Journal of proteome research 16.5 (2017): 2080-2090.

The aim of table 1 was to summarize the most frequently used multi-omics layers, which are relevant in cancer research. After thorough evaluation, although ionomics seems an interesting data type, we found it not extensively used in the field of interest. However, we agree that lipidomics is a valid extension to our collection and therefore we have added it to table 1. We hope reviewer 1 agrees with our conclusions.

3) Section 1.4: I suggest the authors to conclude the section by mentioning online curated platforms that also provide tools to perform standardised multi-omics data analysis, for example Galaxy (https://galaxyproject.eu/citations).

We agree and we have done accordingly.

4) Section 2.1: because the literature mining approach represent most of the novelty of the paper this section needs to be strengthend. I invite the authors to i) extend the text by adding also here a description of the mining procedure in Supplementary Methods, and ii) to add a multiple panel Figure to describe the statistic of their mining. This main text figure should include Figure S2 and also the second order statistic describing co-occurrence of the selected keywords. 

We agree that the literature mining process is a main part of the manuscript. Therefore, we extended this part in the results section and added an additional Figure 1 as well as an additional supplementary Figure S3. 

5) Section 2.2.2: expand description. The enumerated tools integrate single mic data to extract multi-omics markers? Where is the multi-comic aspect of these tools?

We have expanded the description to clarify the biomarker discovery section and added citations. 

6) Section 2.2.4: Drug analysis, again I invite the authors to expand the text and highlight the multi-comic aspects of the tools within this analysis category.

Again, we have expanded the description and added a multi-omics study as an example.

7) Figure 1: Quality too low. Authors must provide high quality figures. Additionally they have to better describe what they show in the caption (MOFA is mentioned at the third line of caption!) 

8) Figure 2: Quality too low. Unreadable. Authors must provide high quality figures. 

9) Figure 3: Quality too low. Unreadable. Authors must provide high quality figures. 

10) Figure 4: Quality too low. Unreadable. Authors must provide high quality figures. 

11) Figure 4: Quality too low. Authors must provide high quality figures.  

Thank you for this comment. Although the original figures were of good quality, the manuscript insertions were blurry. This has been fixed. Also the caption of figure 2 (formerly 1) has been expanded for better comprehensiveness

Other comments 

12) Line 67 : "the vast ... as an iceberg"; reformulate the sentence. It doesn't sound good.

This expression has been adjusted.

13) Line 70 : perception -> understanding ? What do the authors mean by "reflecting the perturbed components restricted to relative molecular layer?

This has been corrected/clarified.

14) Line 92: worth citing here also Orlandi, Ester, et al. "Potential role of microbiome in oncogenesis, outcome prediction and therapeutic targeting for head and neck cancer." Oral oncology 99 (2019), since the authors included microbiome in Table 1. 

Done.

15) Line 243: in addition to reference 43 I strongly encourage to cite here i) Haas, Robert, et al. "Designing and interpreting ‘multi-omic’experiments that may change our understanding of biology." Current Opinion in Systems Biology 6 (2017), and ii) Iacovacci, Jacopo, et al. "Extraction and Integration of Genetic Networks from Short-Profile Omic Data Sets." Metabolites 10.11 (2020).

The suggested citations have been added.

16) Lines 248-249 : "standardized" protocol; the need for standardised analysis is an important point, please explain the concept of standardisation to facilitate readability. "Under-utilization" -> underutilization, is one word without quotes.

This has been expanded on.

17) Line 305: in addition to reference 77 and 78 worth citing here the following review : Simidjievski, Nikola, et al. "Variational autoencoders for cancer data integration: design principles and computational practice." Frontiers in genetics 10 (2019).

We agree. This reference has been added.

18) Line 402-403 : dicovery -> discovery; marker -> markers.

This has been corrected.

19) Table 7: caption unclear.

The table header has been clarified.

20) Line 614: the main novelty of our study lies in the implementation of two reproducible cases. Meaning unclear, explain better. 

We hope that with our added text we have clarified the meaning of this section.

21) Line 623: "we can robustly summarize and highlight ... data integration". I don't understand this phrase.

We have rephrased this sentence to improve readability.

22) Line 707: "General limitations ...'. I would add a couple of lines describing a concrete example. My suggestion: the SIMMS (Subnetwork Integration for Multi-Modal Signatures, Haider, Syed, et al. "Pathway-based subnetworks enable cross-disease biomarker discovery." Nature communications 9.1 (2018)), is a well known multi-omics pathway analysis tool in cancer research, but because it is describe by the word "multi-modal" and not "multi-omics" it wasn't detected.   

We added this paper as an example of literature, which we have missed because of the chosen search terms. 

23) Line 721 and 726: why bullets points 6. And 7. ?

This has been corrected.

Reviewer 2 Report

General comments:

The authors describe this paper as being a review of current computational tools for integration of various data within cancer research. They also show how to use two specific tools on public cancer omics data.

The authors cover an interesting and relevant topic, but they fall short in presenting a clear argument about the content of the paper.

The major concern is that, the manuscript started as a review of many existing tools and methods, but then it jumps to using two specific tools without any justification on why those where chosen. While the use cases presented were interesting, the authors do not compare the performances/results (or any other metric) of those tools with all the other mentioned. If the authors wants to present those specific tools as useful in cancer research, they should just focus on those and reduce the claim that the paper is a review of current computational tools. If the authors wants to present a review of current computational tools, they should more clearly state (with comparison data) why they pick those two specific to highlight.
While it is reasonable that is the researcher responsibility to decide which tools/methods are the most appropriate to their data, it is the authors responsibility to show how tools compares, especially if they claim to provide a comprehensive review.

Another very important aspect is that the authors need to go through the entire text and tone down the claims that they are including "all" the tools available and that this is a "comprehensive and detailed catalogue" of recent tools. The authors are starting with a very specific set of queries, in a very specific database, subsetting with a very specific set of synonyms. While it is understandable that the review needs to have parameters that limit the included results, there are several instances where the authors claim to be more comprehensive than what they actually can be.

Some additional comments about specific sections of the paper are listed below.

Outstanding major issues:

- The word "Evaluation" in the title should be removed or changed as it is misleading. The paper contains no specific benchmark evaluation of tools/methods against each other, and while it provides a useful review of some exiting tools, the presented content is a list of tools resulting from a query followed by two selected examples. The word "evaluation" implies some kind of comparison on different tools with the same benchmark data, not just listing.

- The authors mention that their work is aimed at "creating a framework" for integration of various data in cancer research, but the manuscript only contains two examples that use two different tools and this does not constitute a "framework" that is general. Please rephrase these concepts, or consider focusing the content on the strength/weaknesses of the selected tools rather than the idea of a comprehensive review.

- Similarly to the previous point, the authors claim that what they are presenting will be "facilitating the appropriate selection" of tools for data analysis. The authors then fail to outline what are the selection criteria, what are the considerations that researchers should use to select, and they don't offer any comparison measure between the different tools. How is the reader supposed to use the information to make an informed appropriate selection?

- In the initial sections, Bioconductor and R are mentioned, but no mention of other platforms/languages (like Python, Java and Matlab) is included. Some of the tools listed later in the tables use those frameworks so if the authors want to provide a comprehensive review, they should consider adding information about all those different frameworks.

- The jump between the list of tools in the various tables and the sections with the MOFA/netDX examples is too abrupt. The authors do not explain why they picked those examples, and why they picked those tools. Reading from section 2.2 to 2.3/2.4 the reader is not guided into the discussion points, but just shown results in figures with information that is neither explained nor contextualized.

- The jump between section 2.3 to section 2.4 is also very abrupt. Why is a different tool used? why a different dataset? The readers need to be guided along the various sections in a logical way if the authors want this to be a useful review.

Other issues identified:
- reference 130 is available on biorxiv, but the biorxiv source is not shown in the reference.
- in table 7 it is odd that MOFA has 3 citations while all other tools/dataset have only one. Please be consistent.
- "multi-omics data could stably identify links" --> please remove the word "stably".
- The layout of tables (e.g., Table 1) with center alignment of text makes it very hard to read. Please consider left align the text in each cell and add some shading of the rows to improve readability.
- page 3, line 115, typo, missing space "Consortiumwas".
- page 5, line 195, the sentence seems to be incomplete/cut.
- lines 334-337 do not flow very well, please rephrase to improve readability.
- please find a synonym for the word "robustified".
- often authors use "research aim" where it could be more appropriate to use the expression "research goal" .
- the authors mentioned some tasks were executed "on a Windows 10 laptop". It would be more appropriate to be more specific and rigorous in the description of the hardware/software (and memory/time) requirements for the data analysis.

Author Response

General comments:

The authors describe this paper as being a review of current computational tools for integration of various data within cancer research. They also show how to use two specific tools on public cancer omics data.

The authors cover an interesting and relevant topic, but they fall short in presenting a clear argument about the content of the paper.

The major concern is that, the manuscript started as a review of many existing tools and methods, but then it jumps to using two specific tools without any justification on why those where chosen. While the use cases presented were interesting, the authors do not compare the performances/results (or any other metric) of those tools with all the other mentioned. If the authors wants to present those specific tools as useful in cancer research, they should just focus on those and reduce the claim that the paper is a review of current computational tools. If the authors wants to present a review of current computational tools, they should more clearly state (with comparison data) why they pick those two specific to highlight.
While it is reasonable that is the researcher responsibility to decide which tools/methods are the most appropriate to their data, it is the authors responsibility to show how tools compares, especially if they claim to provide a comprehensive review.

Another very important aspect is that the authors need to go through the entire text and tone down the claims that they are including "all" the tools available and that this is a "comprehensive and detailed catalogue" of recent tools. The authors are starting with a very specific set of queries, in a very specific database, subsetting with a very specific set of synonyms. While it is understandable that the review needs to have parameters that limit the included results, there are several instances where the authors claim to be more comprehensive than what they actually can be.

We agree with you that our work is far from being comprehensive and we have toned down our claims accordingly. However, we tried different combinations of keywords to cover as many relevant publications as possible. Additionally, we added Table 8 to more clearly describe our finally search strategy.

Some additional comments about specific sections of the paper are listed below.

Outstanding major issues:

- The word "Evaluation" in the title should be removed or changed as it is misleading. The paper contains no specific benchmark evaluation of tools/methods against each other, and while it provides a useful review of some exiting tools, the presented content is a list of tools resulting from a query followed by two selected examples. The word "evaluation" implies some kind of comparison on different tools with the same benchmark data, not just listing.

We thank the reviewer for pointing this out, and we agree. Therefore, we changed the title to reflect the reviewer’s concerns to ‘A detailed catalogue of multi-omics integration methodologies for …’

- The authors mention that their work is aimed at "creating a framework" for integration of various data in cancer research, but the manuscript only contains two examples that use two different tools and this does not constitute a "framework" that is general. Please rephrase these concepts, or consider focusing the content on the strength/weaknesses of the selected tools rather than the idea of a comprehensive review.

We agree with this comment and have therefore refrained from using the term ‘framework’ in the revised document. We also reformulated accordingly the discussion section and made some changes to improve readability and understanding of our rationale.

- Similarly to the previous point, the authors claim that what they are presenting will be "facilitating the appropriate selection" of tools for data analysis. The authors then fail to outline what are the selection criteria, what are the considerations that researchers should use to select, and they don't offer any comparison measure between the different tools. How is the reader supposed to use the information to make an informed appropriate selection?

We thank the reviewer for his remarks and have added a paragraph suggesting tool selection criteria to section 2 after table 7.

IN THE DISCUSSION

- In the initial sections, Bioconductor and R are mentioned, but no mention of other platforms/languages (like Python, Java and Matlab) is included. Some of the tools listed later in the tables use those frameworks so if the authors want to provide a comprehensive review, they should consider adding information about all those different frameworks.

The Bioconductor/R section of our manuscript serves the purpose of highlight tools for the efficient retrieval and handling of publicly available multi-omics data. As to our knowledge non such tools exist in python, java and matlab, to elaborate on these platforms/languages seems out of scope here.

- The jump between the list of tools in the various tables and the sections with the MOFA/netDX examples is too abrupt. The authors do not explain why they picked those examples, and why they picked those tools. Reading from section 2.2 to 2.3/2.4 the reader is not guided into the discussion points, but just shown results in figures with information that is neither explained nor contextualized.

We agree that our rationale for choosing the used tools and datasets was not well phrased out. We have changed this by adding a new section called ‘Rationale for selection of tools and datasets’.

- The jump between section 2.3 to section 2.4 is also very abrupt. Why is a different tool used? why a different dataset? The readers need to be guided along the various sections in a logical way if the authors want this to be a useful review.

As we have added a new section before the MOFA and netDX sections, which extensively discusses our rationale, we believe the jump from MOFA to netDX will not feel so abrupt anymore.

Other issues identified:
- reference 130 is available on biorxiv, but the biorxiv source is not shown in the reference.

Unfortunately, it is not possible to show the biorxiv source, due to the fact that the specific reference layout suggested from the journal does not display it through the EndNote reference manager.

- in table 7 it is odd that MOFA has 3 citations while all other tools/dataset have only one. Please be consistent.

This has been corrected.

- "multi-omics data could stably identify links" --> please remove the word "stably".

Done.

- The layout of tables (e.g., Table 1) with center alignment of text makes it very hard to read. Please consider left align the text in each cell and add some shading of the rows to improve readability.

We changed the alignment for table 1 which indeed increased readability. However, we felt that the other tables were more readable with center alignment and were thus left as is.

- page 3, line 115, typo, missing space "Consortiumwas".

Corrected.

- page 5, line 195, the sentence seems to be incomplete/cut.

Corrected.

- lines 334-337 do not flow very well, please rephrase to improve readability.

Done.

- please find a synonym for the word "robustified".

Has been changed to ‘improved’.

- often authors use "research aim" where it could be more appropriate to use the expression "research goal" .

We took this under consideration and agreed with the suggested wording.

- the authors mentioned some tasks were executed "on a Windows 10 laptop". It would be more appropriate to be more specific and rigorous in the description of the hardware/software (and memory/time) requirements for the data analysis.

We have added more detailed specifications of the machine used for the analysis to enable a better evaluation of system requirements for this type of analysis.

Round 2

Reviewer 1 Report

The authors have addressed my comments in a satisfactory manner. I see that for some figures the quality of the figures might be improved. I recommend the paper for publication in the present form.

Author Response

Dear Reviewer,

thank you very much for your comments and than you find satisfactory all the updates that we included in our revised manuscript, based on your fruitful suggestions. Regarding the figures, we made some further improvements in our final revised document, as we have also uploaded separately each figure with high resolution. Thank you one more time for your very helpful and crucial comments that led to a significant improvement of our review.

With Kind Regards,

The authors
